# The Ync13–Rga7–Rng10 complex selectively coordinates secretory vesicle trafficking and secondary septum formation during cytokinesis

Sha Zhang[1☉], Davinder Singh[1☉], Yi-Hua Zhu[1], Katherine J. Zhang[1], Alejandro Melero[2], Sophie G. Martin[2,3], Jian-Qiu Wu[1]*

1 Department of Molecular Genetics, The Ohio State University, Columbus, Ohio, United States of America, 2 Department of Fundamental Microbiology, University of Lausanne, Lausanne, Switzerland, 3 Department of Molecular and Cellular Biology, University of Geneva, Geneva, Switzerland

☉ These authors contributed equally to this work.
* wu.620@osu.edu

## Abstract

Cytokinesis requires precise coordination of contractile-ring constriction, vesicle trafficking and fusion to the plasma membrane, and extracellular matrix assembly/remodeling at the cleavage furrow to ensure faithful cell division and maintain cell integrity. These processes and proteins involved are broadly conserved across eukaryotes, yet molecular mechanisms controlling the spatiotemporal pathways of membrane trafficking remain poorly understood. Here, using fission yeast genetics, microscopy, and in vitro binding assays, we identify a conserved module including the Munc13 protein Ync13, F-BAR protein Rga7, and coiled-coil protein Rng10 to be critical for precise and selective vesicle targeting to the plasma membrane during cytokinesis. The module specifically recruits the TRAPP-II but not the exocyst complex to tether vesicles containing the glucan synthases Bgs4 and Ags1 along the cleavage furrow. Ync13 subsequently interacts with the SM protein Sec1 for vesicle fusion. Mutations in this pathway disrupt septum integrity and lead to cell lysis. Our work provides key insights into how membrane trafficking is tightly controlled to maintain cell integrity during cytokinesis.

## Introduction

Cytokinesis is a highly conserved cellular process occurring at the late stage of the cell division cycle, resulting in the generation of two daughter cells. From fungi to humans, this process involves coordinated actions of division site selection, actomyosin contractile ring assembly and constriction, plasma membrane deposition and invagination at the cleavage furrow, and extracellular matrix formation or remodeling [1–6]. In fungal cells, including the fission yeast *Schizosaccharomyces pombe*, two layers of new plasma membranes and a division septum form behind the constricting

**Data availability statement:** All relevant data are within the paper and its Supporting Information files.

**Funding:** The work was supported by the National Institute of General Medical Sciences of the National Institutes of Health grants R01 GM118746 and GM118746-06S1 to J-QW and by Swiss National Science Foundation grant 310030_191990 to SGM. J-QW was supported by fellowships from the Swiss National Science Foundation (grant ISZEZ0_200027) and Herbette Foundation at UNIL to SGM during his faculty professional leave. SZ was supported by the Pelotonia Postdoctoral Fellowship Program. Y-HZ was supported by the Pelotonia Graduate Fellowship Program. KJZ was supported by the Pelotonia Undergraduate Fellowship Program. The funders had no role in the study design, data collection and analysis, decision to publish, or preparation of the manuscript.

**Competing interests:** The authors have declared that no competing interests exist.

**Abbreviations:** Co-IP, co-immunoprecipitation; EMCCD, electron-multiplying charge-coupled device; FL, full length; FPS, frames per second; FWHM, Full Width at Half Maximum; ROI, region of interest; SM, Sec1/Munc18; TRAPP-II, Transport Particle Protein-II; WT, wild type.

ring during cytokinesis. The septum is a three-layered cell wall structure that consists of a middle primary septum flanked by secondary septa on each side.

The septum is primarily constructed by three essential transmembrane glucan synthases. The β-Glucan synthase Bgs1/Cps1 synthesizes the linear (1,3)β-glucan for the primary septum [7–11], while the β-glucan synthase Bgs4/Cwg1 and the α-glucan synthase Ags1/Mok1 are mainly involved in the formation of the secondary septum [9,12–15]. Once the septum is complete and mature, daughter cells separate via digesting the primary septum mainly by the glucanases Eng1 and Agn1 [16–18]. Proper plasma membrane deposition, septum formation, and daughter-cell separation are essential for maintaining cell integrity and viability during cytokinesis, especially due to the high cellular turgor pressure in fungal cells [19–22]. The F-BAR protein Cdc15, transmembrane protein Sbg1, Rho1 GTPase, and other proteins help recruit and/or activate Bgs1 behind the contractile ring to build the primary septum [10,23–31]. Cdc15 binds to the plasma membrane through its F-BAR domain and helps deliver Bgs1 to the plasma membrane while the paxillin Pxl1 mediates the interaction of Bgs1 to the contractile ring [25,32,33]. Septins, the anillin Mid2, the exocyst complex, the Rho3 and Rho4 GTPases, and Rho4 GEF Gef3 concentrate the glucanases Eng1 and Agn1 to the rim of the division plane for daughter-cell separation [34–42]. In addition, the Transport Particle Protein-II (TRAPP-II) complex is also important for Eng1's localization at the centroid of the division plane [6,10,23–29]. However, the mechanisms that control the precise spatiotemporal localizations of Bgs4 and Ags1 on the plasma membrane at the division plane for secondary septum formation remain poorly understood.

The coordination of exocytosis and endocytosis is essential for successful cytokinesis. Exocytosis delivers the proteins and membranes needed for furrow ingression and extracellular matrix (including cell wall) formation or remodeling to the division site [16–18,43]. Exocytosis, a highly regulated process, involves several sequential stages: vesicle delivery/trafficking, tethering and docking onto the plasma membrane, priming of the fusion machinery, and membrane fusion via SNARE complex assembly [5,44–48]. In fission yeast cytokinesis, the TRAPP-II complex recognizes and tethers secretory vesicles along the whole cleavage furrow, while the octameric tethering complex exocyst mainly tethers vesicles that contain the glucanases Eng1 and Agn1 and other cargos at the rim of the division plane although it also functions along the furrow [6,42,49]. During mammalian synaptic exocytosis, after vesicle tethering, three SNARE proteins—synaptobrevin-2 (Syb2) on synaptic vesicles, and syntaxin-1 (Syx1) and SNAP-25 (SN25) on the plasma membrane form a ternary *trans*-SNARE complex [50–52]. This complex brings the vesicle and plasma membrane into close proximity to facilitate membrane fusion. Vesicle tethering and fusion are regulated by Rab GTPases and several other proteins including: the Sec1/Munc18 (SM) family protein Munc18-1, which initially locks Syx1 in a closed conformation to inhibit SNARE complex assembly; and the Munc13/UNC-13 family protein Munc13-1, which catalyzes the transition from the Munc18-1/Syx1 complex to the SNARE complex in the presence of SN25 and Syb2 [53–57]. In *S. pombe*, Ync13 is the homolog of Munc13 and UNC-13. Ync13 localizes to cell tips during interphase and the plasma

membrane at the cleavage furrow during cytokinesis [58]. Deletion of Ync13 (ync13Δ) results in defective exocytosis, impaired endocytosis, uneven distribution of cell wall enzymes at the division site, and extensive cell lysis during cell separation [58]. However, the regulatory mechanisms of Ync13 in membrane dynamics, its binding partners, and the exact cause of cell lysis upon its deletion remain unclear.

Endocytosis also plays essential and dynamic roles in cytokinesis, particularly in membrane remodeling, signal transduction, and recycling membrane proteins. Cells internalize membrane lipids and proteins from the plasma membrane mainly through the clathrin-mediated endocytosis from the growth and division sites [59–65]. During cytokinesis, endocytosis helps retrieve old/inactive or excess membrane proteins and recycle materials to be reused for septum formation or membrane expansion [5,61,63,66]. Thus, endocytosis works hand-in-hand with exocytosis to maintain the proper levels of lipids, glucan synthases, and other proteins needed for plasma membrane deposition and septum formation. Yeast endocytosis is actin-dependent and many endocytic mutants in genes such as the endocytic adaptor ede1, fimbrin fim1, and clathrin light chain clc1 result in furrow ingression defects, cell wall abnormalities, or cytokinesis failure [59,67–73].

Rga7 is a Rho2 GAP in fission yeast that plays crucial roles in cytokinesis [73–76]. It contains an F-BAR domain necessary for its membrane association and a Rho GAP domain at its C-terminus. Rga7 localizes to the plasma membrane at the cell tips during interphase and relocates to the division site during cytokinesis, a process dependent on its interaction with the coiled-coil protein Rng10 and membrane lipids [73–76]. Without Rng10, Rga7 essentially disappears from the division site, leading to defective septum formation and cell lysis [74,75]. Rga7 collaborates with F-BAR proteins Cdc15 and Imp2, C2 domain-containing protein Fic1, and the paxillin Pxl1 to maintain actomyosin ring stability and ensure successful septum formation and separation [32,73,77–79]. Similar to Ync13, Rga7 and Rng10 collaboratively regulate the accumulation and dynamics of glucan synthases [74,75]. Rga7 is reported to facilitate the trafficking of β-glucan synthase Bgs4 from the Golgi to the plasma membrane [76]. Cells lacking Rga7 show defects similar to those observed in bgs4 and ync13 mutant cells [74,75]. However, the relationships between Ync13, Rga7–Rng10, the TRAPP-II complex, and Sec1 in cytokinesis were unknown.

In this study, we aimed to map out some key spatiotemporal pathways for plasma membrane deposition and septum formation during cytokinesis using fission yeast as a model system by mistargeting proteins to mitochondria, co-immunoprecipitation, in vitro binding assays, genetic and cellular methods, live-cell confocal microscopy, and electron microscopy. We find that Ync13 regulates exocytosis by interacting with Rga7–Rng10, the SM protein Sec1, and the TRAPP-II complex. The Ync13–Rga7–Rng10 module selectively controls the accumulation and distribution of glucan synthases involved in secondary septum formation, such as Bgs4 and Ags1; rather than Bgs1, which is essential for the primary septum assembly. Consistently, the secondary septum is defective in ync13 mutants, which leads to cell lysis during daughter-cell separation. Collectively, we find that the Ync13–Rga7–Rng10 module and the TRAPP-II complex are central players for plasma membrane dynamics and secondary septum formation during cytokinesis.

## Results

### Mapping the physical interactions among the key cytokinetic proteins involved in plasma membrane deposition and septum formation by ectopic mistargeting

Unlike the proteins in the contractile ring and its precursor nodes [80–84], the physical interactions and functional relationships among the proteins involved in plasma membrane deposition/expansion and septum formation are poorly understood. These proteins are crucial for plasma membrane and septum integrity, exocytosis, and endocytosis. In this study, we used the fission yeast S. pombe as a model system to test the interactions and functional relationships of the key proteins in these processes. We first mistargeted mEGFP, GFP, or mECitrine tagged proteins to mitochondria using outer mitochondrial membrane protein Tom20 tagged with GFP-binding protein (GBP) nanobody, then examined if tdTomato, mCherry, or RFP tagged proteins were also mistargeted to mitochondria (see Table 1 for summary). All the tagged genes are the sole copies of the genes in the cells. Our and others' previous studies have shown this strategy is highly efficient

**Table 1. Summary of Tom20-GBP mistargeting assays.**

| Protein 1 | Protein 2 | Protein 2 mislocalized to mitochondria (Yes/No)? | Figure |
|---|---|---|---|
| Rga7-mEGFP | Ync13-tdTomato | Y | Fig 1B |
| Rng10-mEGFP | Ync13-tdTomato | Y | Fig 1B |
| Rga7-mEGFP | Trs120-tdTomato | Y | Fig 1C |
| Rng10-mEGFP | Trs120-tdTomato | Y | Fig 1C |
| Rga7-mEGFP | RFP-Bgs4 | Y | Fig 1D |
| Rng10-mEGFP | RFP-Bgs4 | Y | Fig 1D |
| Rga7-mEGFP | Ags1-mCherry | Y | Fig 1E |
| Rng10-mEGFP | Ags1-mCherry | Y | Fig 1E |
| Rga7-mEGFP | Smi1-tdTomato | Y | Fig 1F |
| 3nmt1-mECitrine-Ync13 | Rga7-mCherry | Y | S2A Fig. |
| 3nmt1-mECitrine-Ync13 | Rng10-mCherry | Y | S2A Fig. |
| 3nmt1-mECitrine-Ync13 | Sec1-tdTomato | Y | S2B Fig. |
| 3nmt1-mECitrine-Ync13 *rng10Δ* | Sec1-tdTomato | Y | S2G Fig. |
| 3nmt1-mECitrine-Ync13 | RFP-Bgs4 | Y | S3B Fig. |
| 3nmt1-mECitrine-Ync13 | Ags1-mCherry | Y | S3B Fig. |
| Rga7-FBD-mEGFP | Ync13-tdTomato | Y | S4B Fig. |
| Rga7(ΔF-BAR)-mEGFP | Ync13-tdTomato | Y | S4C Fig. |
| mECitrine-Rng10-(201-1038) | Ync13-tdTomato | Y | S4E Fig. |
| Rng10(1-750)-mEGFP | Ync13-tdTomato | Y | S4F Fig. |
| mECitrine-Rng10-(751-1038) | Ync13-tdTomato | Y | S4G Fig. |
| 3nmt1-mECitrine-Ync13 | Sec3-mCherry | N | S2C Fig. |
| 3nmt1-mECitrine-Ync13 | Ede1-mCherry | N | S2D Fig. |
| 3nmt1-mECitrine-Ync13 | Fim1-mCherry | N | S2E Fig. |
| 3nmt1-mECitrine-Ync13 | Clc1-mCherry | N | S2F Fig. |
| Rga7-mEGFP | Sec3-tdTomato | N | S3A Fig. |
| Rng10-mEGFP | Sec3-tdTomato | N | S3A Fig. |
| Rga7-mEGFP | tdTomato-Bgs1 | N | S3C Fig. |
| Rng10-mEGFP | tdTomato-Bgs1 | N | S3C Fig. |
| Rng10-(1-200)-mEGFP | Ync13-tdTomato | N | S4D Fig. |
| mECitrine-Rng10-(751-1038) *rga7Δ* | Ync13-tdTomato | N | S4H Fig. |

at detecting protein physical interactions [6,13,47,74,75,85]. As shown previously [86,87], GBP does not bind to tdTomato, mCherry, or RFP; and no signal bleed through between the green/yellow and red channels were detected (S1 Fig).

We started with Ync13, Rga7, and Rng10 because they perfectly colocalized on the plasma membrane at cell tips and the division site (Fig 1A), and their mutations lead to cell lysis during daughter-cell separation [58,74,75]. We first mistargeted Rga7-mEGFP or Rng10-mEGFP to mitochondria using Tom20-GBP. The ectopically targeted Rga7 and Rng10 were both able to recruit Ync13-tdTomato to the mitochondria (Fig 1B and Table 1). We detected mitochondrial localization of Ync13 in all Tom20-GBP Rga7-mEGFP ($n$ = 120 cells) and Tom20-GBP Rng10-mEGFP expressing cells ($n$ = 200 cells), except in some unhealthy cells that Ync13 diffused in the whole cytoplasm, which could be due to the side effects (besides cell lysis which is obvious in the DIC channel) of mislocalization of Rga7, Rng10, and Ync13 (Fig 1B). Mistargated proteins concentrated into clusters or linear structures instead of over the whole mitochondria (Fig 1B, examples marked by arrowheads). In addition, the ectopically targeted mECitrine-Ync13 recruited the Rga7 and Rng10 to the mitochondria (S2A Fig). In this assay, Ync13 was overexpressed using the *3nmt1* promoter under repressed condition

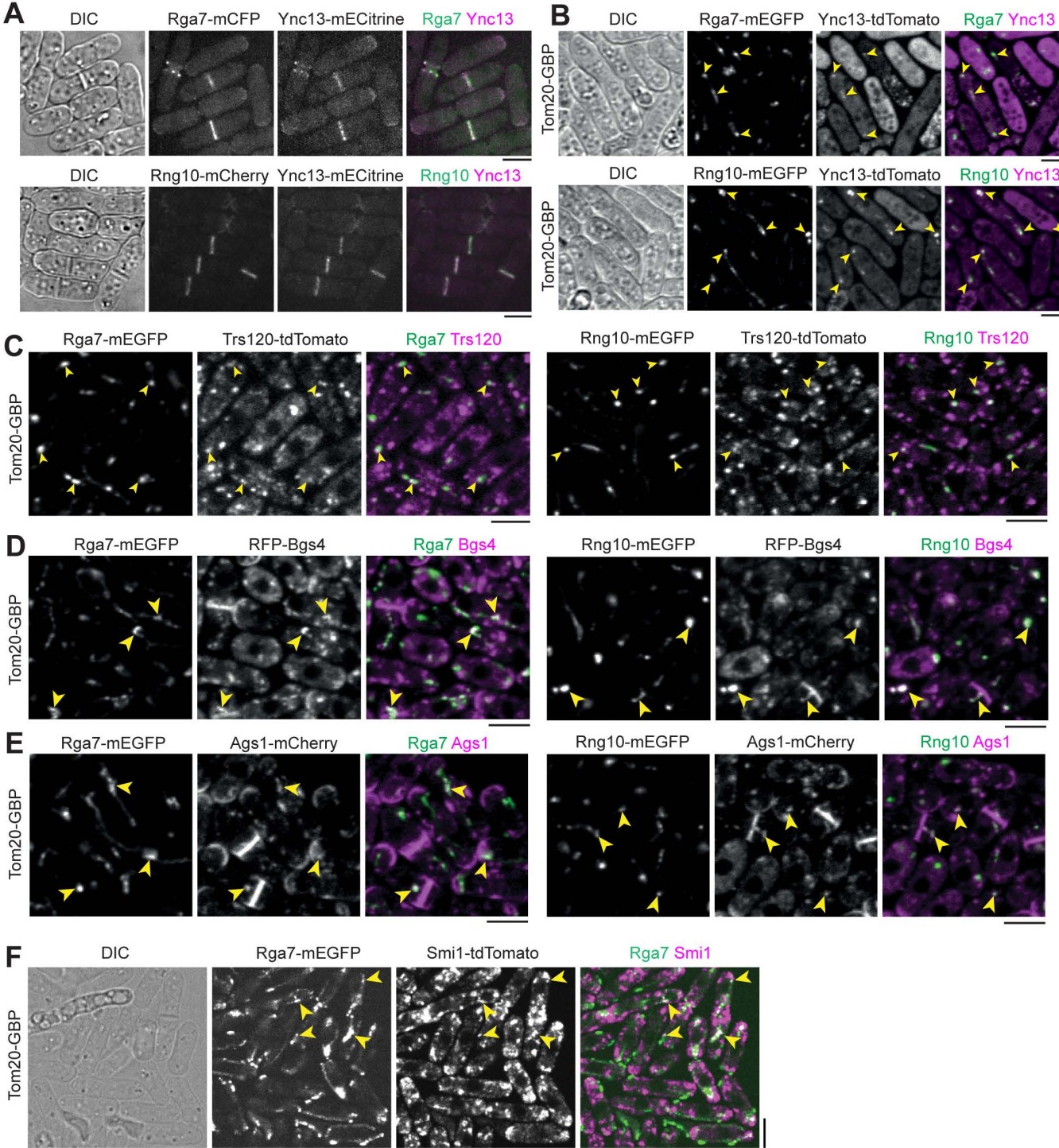

**Fig 1. Physical interactions among the key cytokinetic proteins in plasma membrane deposition and septum formation revealed by ectopic mistargeting to mitochondria by Tom20-GBP.** Arrowheads mark examples of colocalization at mitochondria. **(A)** Ync13 colocalizes with Rga7 and Rng10 at cell tips and the division site. **(B–F)** Tom20-GBP can ectopically mistarget Rga7/Rng10-mEGFP and their interacting partners tagged with tdTomato/RFP/mCherry to mitochondria. Tom20-GBP was used to recruit mEGFP-tagged Rga7 or Rng10 to mitochondria, and colocalization was assessed with tdTomato/RFP/mCherry-tagged candidate binding partners. Cells were grown at 25ºC in YE5S + 1.2 M sorbitol medium for ~36–48 h and then were washed with YE5S without sorbitol and grown in YE5S for 4 h before imaging. **(B)** Rga7/Rng10 and Ync13. **(C)** Rga7/Rng10 and Trs120. **(D)** Rga7/Rng10 and Bgs4. **(E)** Rga7/Rng10 and Ags1. **(F)** Rga7 andSmi1. Bars, 5 μm.

(in medium with thiamine) because Ync13's native level was too low to spread sufficiently to the abundant Tom20-GBP. These data suggested that Ync13 physically interacts with both Rga7 and Rng10.

Because *ync13* mutants are defective in both exocytosis and endocytosis [58], we tested if Ync13 could mistarget key proteins in exocytic and endocytic pathways to mitochondria by Tom20-GBP. The ectopically targeted mECitrine-Ync13 was able to recruit the Sec1/Munc18 (SM) family protein Sec1-tdTomato to the mitochondria (S2B Fig), suggesting Ync13 may function as a priming factor for SNARE complex assembly and/or vesicle tether during exocytosis similar to its animal homologs Munc13/UNC-13 [53–55,88–92]. In addition, Ync13 was still able to mistarget Sec1 to mitochondria without Rng10 (S2G Fig). This suggests that Rng10 (maybe together with Rga7) is not required for the interaction between Ync13 and Sec1. However, Ync13 could not recruit the following proteins to mitochondria: the exocyst subunit Sec3 (S2C Fig); or endocytic proteins early coat marker Eps15 protein Ede1 [68], actin crosslinker fimbrin Fim1 [69,93], and clathrin light chain Clc1 [67] (S2D–S2F Fig). Thus, Ync13 interacts with the SNARE-binding protein Sec1 that is involved in exocytosis, but not with the exocyst or various proteins in endocytosis.

We next tested if Rga7 or Rng10 can mistarget TRAPP-II vesicle tether and secretory cargos to mitochondria by Tom20-GBP. Both Rga7-mEGFP and Rng10-mEGFP recruited tdTomato-tagged Trs120, which is a TRAPP-II-specific subunit [94,95], but not the exocyst subunit Sec3 [6,42,96], to mitochondria (Figs 1C and S3A). These data suggest that Rga7 and Rng10 selectively interact with the TRAPP-II complex to promote vesicle tethering during exocytosis along the cleavage furrow, rather than the exocyst complex, which is more concentrated at the rim of the division plane [6,42,96]. Mistargeted Rga7-mEGFP and Rng10-mEGFP ectopically recruited RFP-Bgs4 to mitochondria in ~100% of Tom20-GBP Rga7-mEGFP or Tom20-GBP Rng10-mEGFP cells (Fig 1D, *n* > 100 cells), and Ags1-mCherry to mitochondria in ~90% of Tom20-GBP Rga7-mEGFP or Tom20-GBP Rng10-mEGFP cells (Fig 1E, *n* > 100 cells). Consistently, Rga7 mistargeted tdTomato-tagged Smi1, which is an adaptor for Bgs4 [13], to mitochondria in cells expressing Tom20-GBP Rga7-mEGFP (Fig 1F). Similarly, mECitrine-Ync13 could mistarget Bgs4 and Ags1 to mitochondria (S3B Fig). Surprisingly, neither ecto-pically targeted Rga7 nor Rng10 recruited Bgs1 to mitochondria (S3C Fig), suggesting the recruiting of the glucan syn-thases by Rga7 and Rng10 are selective.

Ectopic mistargeting of Ync13-tdTomato to mitochondria using Rga7 or Rng10 truncations further supported that Rga7, Rng10, and Ync13 interact with each other (S4 Fig and Table 1). Except Rng10-(1-200) (S4D Fig), all other Rga7 and Rng10 truncations, even the Rng10 C-terminal (751-1038) could mistarget Ync13-tdTomato to mitochondria (S4 Fig), although less efficiently than the FL proteins. To further dissect the interactions among Rga7, Rng10, and Ync13, we tested whether Ync13 can associate with Rng10 independently of Rga7. mECitrine-Rng10(751-1038) failed to recruit Ync13-tdTomato to mitochondria in *rga7Δ* cells (S4H Fig). Based on what we know about Rga7 and Rng10 interactions and localization interdependency [74,75], the data suggest that the Ync13-Rng10 interaction is bridged by Rga7. Collec-tively, these mistargeting data suggested Rng10, Rga7, and Ync13 form a protein complex, which recruits the TRAPP-II complex and Sec1 but not the exocyst to the plasma membrane for vesicle tethering, SNARE complex assembly, and fusion. The main proteins recruited by the Rng10–Rga7–Ync13 complex are secretory vesicles containing cargos such as the glucan synthases Bgs4 (and its adaptor Smi1) and Ags1, but not Bgs1 or the endocytic machinery.

## Rga7 physically interacts with Ync13, Bgs4, and Smi1

We used co-immunoprecipitation (Co-IP) and in vitro binding assays to confirm some of the major interactions revealed by the mistargeting experiments. It is known that Rng10 directly interacts with Rga7 and both proteins interact with mem-brane lipids [74,75]. The F-BAR domain of Rga7 interacts with Rng10 C-terminal amino acids 751-1038 [74,75]. Smi1 interacts with Bgs4 and is important for Bgs4 localization [13]. Most other interactions suggested by the mistargeting assays had not been tested. For Co-IP assays, all the proteins were expressed under their native promoters. Consistent with the mistargeting results (Figs 1 and S2–S4), Rga7-13Myc was coimmunoprecipitated from fission yeast cell extracts by Ync13-mECitrine (Fig 2A), GFP-Bgs4 (Fig 2B), and Smi1-mEGFP (Fig 2C, left). Moreover, Smi1-13Myc was also

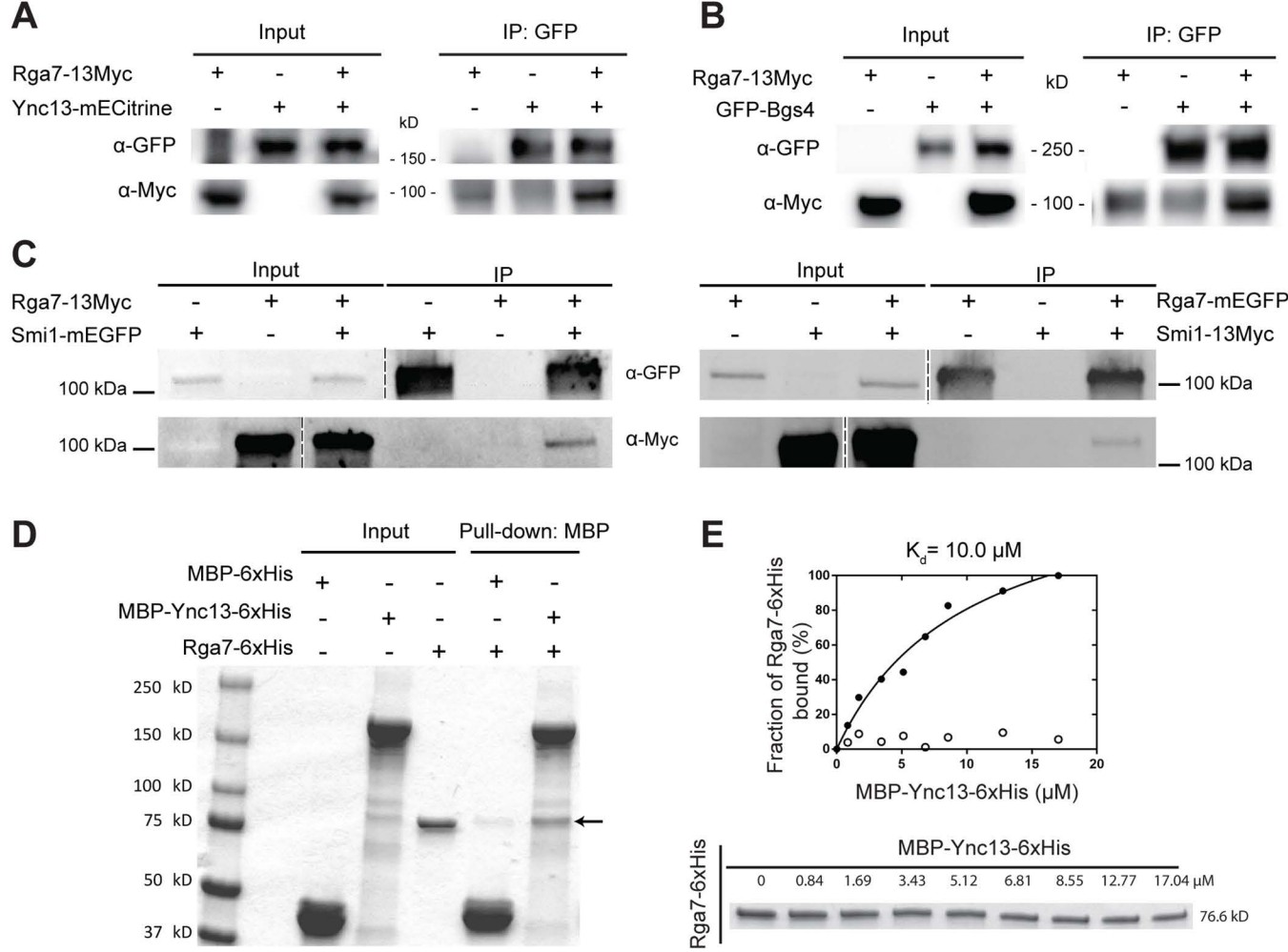

**Fig 2. Confirmation of Rga7's physical interactions with Ync13, Bgs4, Smi1 by co-IP and/or in vitro binding assays. (A-C)** Rga7-13Myc coimmunoprecipitates with Ync13 **(A)**, Bgs4 **(B)**, and Smi1 **(C, left)** from *Schizosaccharomyces pombe* cell extracts using antibodies against GFP. **(C, right)** Smi1-13Myc coimmunoprecipitates with Rga7. The vertical dashed lines in **(C)** mark the positions of protein ladders that were excised out. **(D)** In vitro binding of Rga7 and Ync13 using purified proteins. Bead-bound MBP-Ync13-6xHis or MBP-6xHis control was incubated with Rga7-6xHis. The arrow marks the Rga7-6xHis band. The weak ~76 kDa band in the MBP-6xHis pull-down lane is nonspecific background binding between MBP and Rga7. **(E)** Supernatant depletion assay to measure the $K_d$ between Ync13 and Rga7. (Top) Curve fit showing Rga7 bound fraction (filled circles) vs. MBP-Ync13-6xHis concentration on the beads. Empty circle: MBP-6xHis. (Bottom) Coomassie-stained gel of the supernatants showing Rga7-6xHis depletion. Numbers above each lane indicate total MBP-Ync13-6xHis concentration on the beads. The underlying data (for panel **E**) and uncropped Western blots or SDS–PAGE gels (for panels **A–E**) can be found in S1 Raw Data file and S1 Raw Images.

coimmunoprecipitated by Rga7-mEGFP (Fig 2C, right). Furthermore, when Ync13-3FLAG expressed at its native chromosomal locus and under endogenous promoter was immunoprecipitated from *S. pombe* cell extracts, we identified Rga7 and Rng10 among Ync13's binding partners (although not the top hits) by mass spectrometry analyses (S1 Table). These results suggest that Rga7, Rng10, and Ync13 do form a protein complex, although maybe dynamic and not super stable (see Discussion). Our data indicate that Rga7 interacts with both Ync13 and Rng10 to form a module on the plasma membrane for targeting of vesicles containing cargos such as glucan synthases Bgs4 and Ags1. However, these glucan synthases are multipass integral membrane-embedded proteins and likely only indirectly associate with the module Rng10–Rga7–Ync13, without forming a big protein complex.

Next, we tested if Rga7 and Ync13 directly interact by in vitro binding assays using purified recombinant full-length (FL) Ync13 and Rga7. MBP-Ync13-6His bound Rga7-6His with a dissociation constant ($K_d$) of 10.0 μM (Fig 2D and 2E). These data indicate that Ync13 and Rga7 directly interact. Thus, we conclude that Rng10, Rga7, and Ync13 form a protein complex, which can recruit vesicles containing the glucan synthases Bgs4 (and Smi1) and Ags1 to the plasma membrane at the division plane for secondary septum formation.

To gain structural insight into the organization of the Rga7–Rng10–Ync13 complex, we performed structural modeling using AlphaFold3. Our previous work demonstrated that the F-BAR protein Rga7 forms a stable dimer and its F-BAR domain binds the C-terminal (aa751-1038) region of Rng10 [75]. Based on these findings, we constructed an input model consisting of two FL Rga7 subunits, two Rng10(751–1038) subunits, and one FL Ync13. The predicted structure revealed a modular organization in which Rng10(751–1038) associated strongly with the F-BAR domain of the Rga7 dimer, consistent with our prior biochemical data [75]. In addition, the model suggested that Ync13 interacted with the GAP domain of Rga7, positioning Ync13 in close proximity to the Rga7–Rng10 interface (S5A, S5B, S5D, and S5F Fig). Further domain-specific prediction confirmed the interactions between Rga7-GAP and Ync13 N-terminus aa(1-600) (pTM: 0.63, ipTM: 0.64), Rga7 F-BARs (pTM: 0.74, ipTM: 0.71), as well as Rga7 F-BAR and Rng10(751–1038) (pTM: 0.56, ipTM: 0.78) (S5C–S5F Fig). Overlay analyses revealed that the interacting domains align well with the structure of whole complex as the root mean square differences (RMSDs) are <1 Å for all predictions. In particular, Rga7-GAP domain interacting with Ync13 N-terminus aligned perfectly with predicted structure of whole complex (S5A and S5B Fig). This arrangement provides a structural framework for how the Rga7 dimer, through direct interactions with Rng10 and Ync13, could function as a scaffold to recruit and stabilize these two proteins. Together, these modeling results support our experimental findings and provide a mechanistic rationale for the cooperative function of Rga7, Rng10, and Ync13 at the division site.

## Bgs4 recruitment and distribution at the division plane depends on Rng10, Rga7, and Ync13

Next, we asked the functional significance of the detected physical interactions among the proteins in cytokinesis. We first tested their localization interdependence. We started with Rng10, Rga7, and Ync13 because they colocalized perfectly at cell tips and the division site on the plasma membrane from anaphase until daughter-cell separation (Fig 1A). Compared to wild type (WT), Ync13 levels at the division site measured by fluorescence intensity were significantly reduced (>85%) in both *rga7Δ* and *rng10Δ* cells (Fig 3A and 3B), although the global Ync13 level was not obviously affected (Figs 3C and S6A). We next examined the localizations of Rga7 and Rng10 in *ync13Δ* cells. The total Rga7 amount at the division site increased in *ync13Δ* cells under fluorescence microscopy, although Rga7 and Rng10 global protein levels were not obviously affected in *ync13Δ* cells (slightly increased in *3nmt1-ync13* cells) in Western blotting (Figs 3D, 3E, S6B and S6C). Both Rga7 and Rng10 were significantly more concentrated at the center of division plane in *ync13Δ* cells after the constriction of the contractile ring marked with Rlc1 than in WT cells (Fig 3E–3G). These were confirmed by the calculated Full Width at Half Maximum (FWHM) from Gaussian fits of Rga7 and Rng10 fluorescence intensity across the division site (Fig 3F and 3G). Interestingly, previous work showed that *ync13Δ* leads to similar central accumulation of the glucan synthase Bgs4 at the division site [58], which resembles the aberrant distribution of Rga7 and Rng10 in *ync13Δ* cells (Fig 3E–3G).

Glucan synthases are essential for maintaining septum thickness and integrity during cytokinesis. Cells of *ync13Δ*, *rga7Δ*, *rng10Δ*, *ags1*, and *bgs4* mutants all lyse to different degrees during daughter-cell separation dependent on the growth conditions [9,12,58,74,75]. Because the glucan synthases Ags1 and Bgs4 display lower concentrations at the division site in *rga7Δ* and *rng10Δ* cells [74,75], and Rga7 and Rng10 can mistarget Bgs4 and Ags1 (Fig 1D and 1E), we examined the relationship between mislocalized Rga7, Rng10, Bgs4, and Ags1 in *ync13* mutants. As previously reported [58], compared to WT cells, in *ync13Δ* cells, Ags1 and Bgs4 were more concentrated at cell center during septum maturation (from the end of ring constriction until daughter-cell separation), but Bgs1 was less affected (Fig 4A–4C). We also found that in *ync13Δ* cells, the Bgs4 intensity at the rim of the septum were much lower than in WT after ring constriction

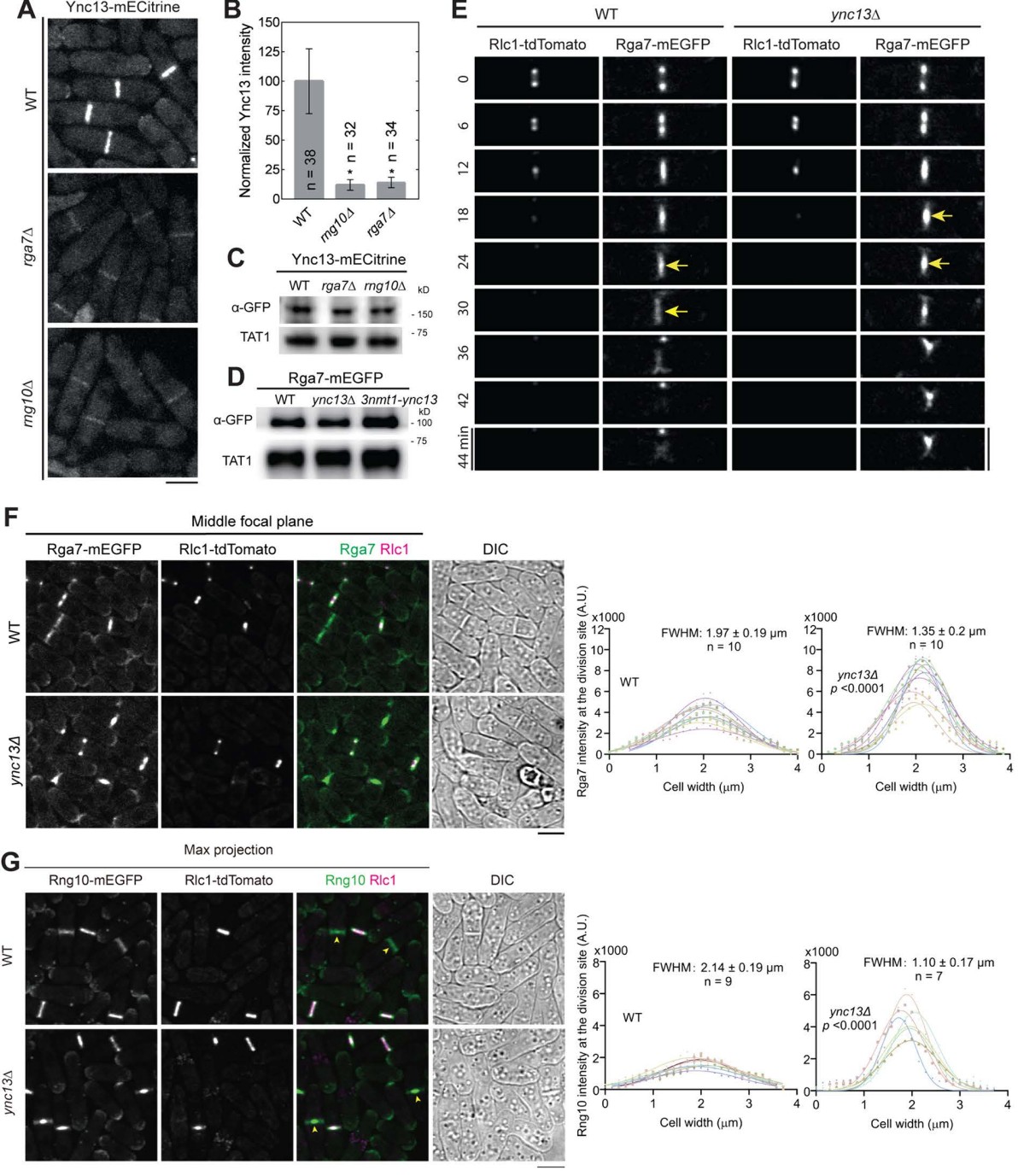

**Fig 3. Interdependence of Ync13, Rga7, and Rng10 localization at the division site. (A and B)** Rng10 and Rga7 are important for Ync13 localization. Micrographs **(A)** and fluorescence intensity at the division site **(B)** of Ync13 in WT, *rga7Δ*, and *rng10Δ*. *p < 0.0001 compared with WT. **(C and D)** Western blotting showing Ync13 **(C)** and Rga7 **(D)** protein levels in cells extracts from the indicated strains. Tubulin detected by the TAT1 antibody was used as a loading control. **(E and F)** Time courses **(E)** and micrographs **(F)** showing Rga7 concentrated to the center of the division site in *ync13Δ* cells during or after ring constriction. Middle focal plane showing Rga7 localization relative to the contractile ring marked with Rlc1. Arrows in **(E)** mark Rga7 localization at the center of the division plane after ring constriction. **(G)** Micrographs showing Rng10 concentrated to the center of the division site in *ync13Δ* cells during late cytokinesis. Max projection and DIC are shown. **(F and G)** Right panels showing Gaussian fits of fluorescence intensity of Rga7 **(F)** and Rng10 **(G)** along the division site during septum maturation (after Rlc1 ring disappearance) in WT or *ync13Δ* cells. FWHM: mean ± standard deviation. Cells were grown exponentially in EMM5S liquid media at 25°C for 48 h in **(E–G)**. The underlying data (for panels **B**, **F**, and **G**) and uncropped Western blots (for panels **C** and **D**) can be found in S1 Raw Data file and S1 raw images. Bars, 5 μm.

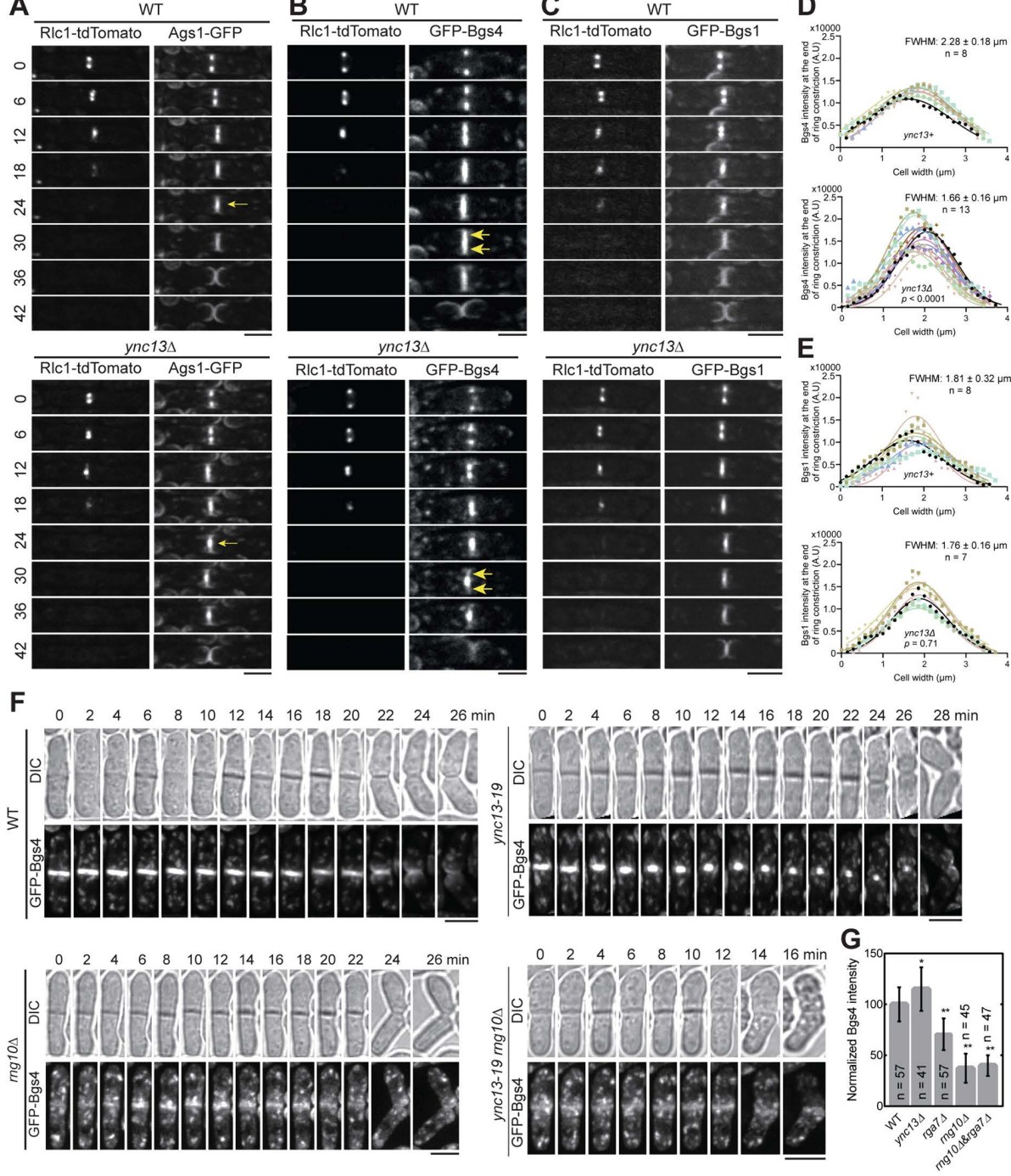

**Fig 4. The spatial distribution of Bgs4 in *ync13Δ* cells depends on Rng10.** Cells were grown exponentially in EMM5S liquid media at 25°C for 48 h before imaging. **(A-C)** Time courses (in min) of Ags1 **(A)**, Bgs4 **(B)**, and Bgs1 **(C)** distribution at middle focal plane along the division site relative to the contractile ring marked with Rlc1 in WT (top) and *ync13Δ* cells (bottom). Arrows in **(A)** mark Ags1 concentrated at the center of the division plane after ring constriction and in **(B)** mark decreased Bgs4 level at the septal edges. **(D and E)** Gaussian fits of fluorescence intensity of Bgs4 **(D)** and Bgs1 **(E)** along the division site at the end of ring constriction in WT or *ync13Δ* cells. FWHM: mean ± standard deviation. **(F)** Time courses of Bgs4 localization in WT, *ync13-19*, *rng10Δ*, and *rng10Δ ync13-19* cells before and during cell separation. Both daughter cells lysed after cell separation in *rng10Δ ync13-19*. Cells were grown exponentially at 25°C and then shifted to 36°C for 2 h before imaging. Bars, 5 μm. **(G)** Normalized Bgs4 fluorescence intensity at the division site during septum maturation in the indicated strains. The underlying data (for panels **D**, **E**, and **G**) can be found in S1 Raw Data file. *$p < 0.001$; and **$p < 0.0001$.

(Fig 4B). In contrast, the distribution of Bgs1 was smoother and the intensity at the edge of the division site in *ync13Δ* cell was similar to WT (Fig 4C). These were confirmed by the calculated FWHM from Gaussian fits of Bgs4 and Bgs1 fluorescence intensity across the division site (Fig 4D and 4E). The lack of Bgs4 at the rim of the division plane may influence the rigidity of secondary septum to counter the internal turgor pressure and cause the daughter-cell lysis during cell separation. Next, we tested if Bgs4 was affected by vesicle tethering and fusion machinery. Consistent with the mistargeting results (Figs 1C and S2B), Bgs4 levels at the division site decreased in *sec1-M2* and *trs120-ts1* mutants after ring constriction (S6D and S6E Fig).

We hypothesize that the *ync13Δ* mislocalizes Rga7 and Rng10, which in turn impacts the distribution of Bgs4 and Ags1 at division site. To test this hypothesis, we first tested Rng10 and Rga7 localization in Bgs4 mutants. Neither Rga7 nor Rng10 localization showed defects in *bgs4* temperature-sensitive mutants *cwg1-1* and *cwg1-2* (S6F and S6G Fig), suggesting that Bgs4 is not important for the localizations of Rga7 and Rng10. Then we compared the localization and intensity of Bgs4 and Bgs1 in WT, *ync13-19*, *rng10Δ*, and *rng10Δ ync13-19* cells at 36°C. The *rga7Δ ync13-19* mutant could not be tested because it was inviable even at 25°C, which is the permissive temperature for *ync13-19* [58]. The time-lapse movies showed that the Bgs4 intensity at division site was significantly reduced in *rng10Δ* and *rng10Δ ync13-19* cells, and Bgs4 did not accumulate in the center of the division plane compared to *ync13-19* cells (Fig 4F). Consistently, Bgs4 levels at the division site were significantly lower in *rga7Δ, rng10Δ,* and *rng10Δ rga7Δ* cells (Fig 4G). In contrast, Bgs1 localization and distribution at the division site remained largely unaffected across all the tested mutants (S7A–S7D Fig). Although line-scan profiles of Bgs1 in the *ync13-19 rng10Δ* double mutant showed a slightly steeper distribution, FWHM analysis revealed no statistically significant difference compared to WT (S7A–S7D Fig). To test whether Bgs1 might display a second wave of recruitment during late cytokinesis, we tracked Bgs1 fluorescence over 2 h movies and aligned intensity profiles at the end of contractile-ring constriction (S7A–S7D Fig). These measurements showed a single peak of Bgs1 intensity at the division site, with no evidence for delayed or secondary accumulation (S7A–S7D Fig). Together, these data indicate Bgs4 is confined at the center of the division plane in *ync13* mutant cells by Rng10 (and Rga7). Thus, Rng10, Rga7, and Ync13 are important for localization, recruitment, and normal spatial distribution of Bgs4, but not Bgs1, at the division site.

## Both Rga7 and Rng10 are involved in TRAPP-II complex-dependent exocytosis at the division site

It is known that Ync13 regulates TRAPP-II-mediated exocytosis [58]. The TRAPP-II complex promotes vesicle tethering for exocytosis along the cleavage furrow and its core subunit Trs120, which is sufficient for TRAPP-II localization, co-localizes with Bgs4-containing vesicles [6,94,95]. Trs120 can mistarget Bgs4 to mitochondria and dynamically localizes to the division site during and after ring constriction [6]. Both *rga7Δ* and *rng10Δ* cells are defective in recruiting glucan synthases to division site and septum formation [73–76]. The *rga7Δ rng10Δ* double mutant is inviable in YE5S rich media at 25°C without osmotic stabilizer sorbitol [74,75]. Therefore, we asked how Rga7 and Rng10 affect the trafficking and tethering of the glucan synthase-containing vesicles, focusing on the TRAPP-II complex because both Rga7 and Rng10 can mistarget Trs120 but not the exocyst subunit Sec3 to mitochondria (Figs 1C and S3A). In *rga7Δ, rng10Δ,* and *rga7Δ rng10Δ* (shifting from medium with sorbitol to the one without sorbitol) cells, Trs120 intensity at division site decreased significantly, while it increased in *ync13Δ* cells (Fig 5A and 5B; arrowheads), consistent with the Bgs4 intensity (Fig 4F and 4G). Trs120 was also more concentrated at the center of the division plane in *ync13Δ* cells (Fig 5A), confirmed the earlier report [58]. The accumulation of Trs120-3GFP outside of the division site in cells with a nearly closed or closed septa was more frequently detected in the sum projection images of 2-min continuous movies in *rga7Δ, rng10Δ,* and *rga7Δ rng10Δ* than in WT cells, suggesting membrane tethering defect at the division site in the mutants (S8A Fig, arrowheads).

We also tracked Trs120 vesicle movement during the late stage of ring constriction. In WT cells, Trs120 puncta were delivered to the cleavage furrow with a bias towards its leading edge near the contractile ring marked by Rlc1 (Fig 5C and 5D; S1 Video). However, in *rng10Δ* cells, Trs120 puncta rarely docked near the leading edge of the division site,

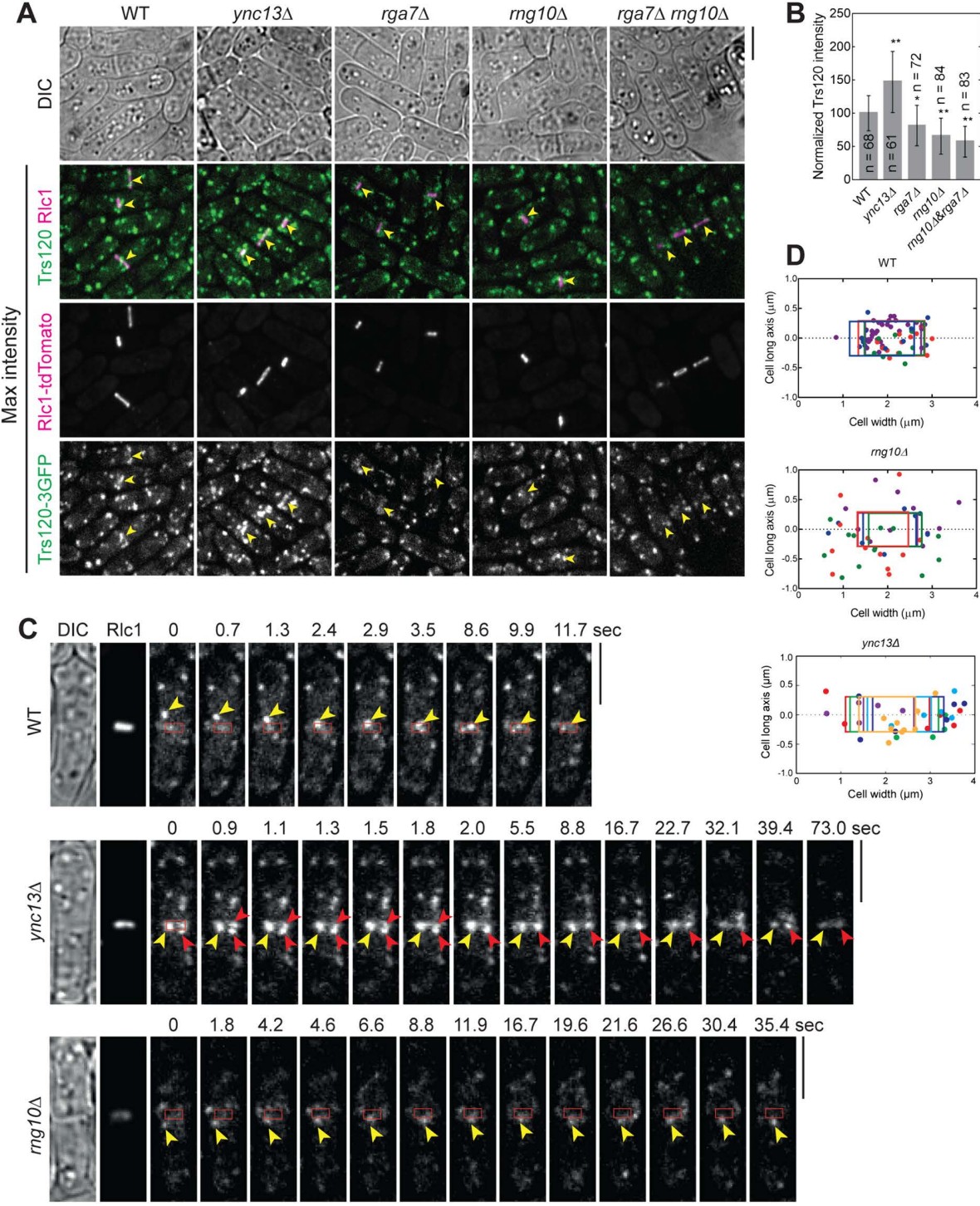

**Fig 5. The Rga7–Rng10–Ync13 module regulates TRAPP-II complex-mediated vesicle tethering and fusion at the division site.** Cells were grown exponentially in EMM5S liquid media at 25°C for ~48 h before imaging. **(A)** Trs120-3GFP localization in WT, *ync13Δ*, *rga7Δ*, *rng10Δ*, and *rga7Δ rng10Δ* cells. Arrowheads mark the Trs120 at division site in cells with constricting ring labeled with Rlc1-tdTomato. **(B)** Quantification of Trs120 intensity at division site in cells with a constricting ring as in (A). *$p < 0.001$ compared with WT, **$p < 0.0001$ compared with WT. **(C)** Trs120 puncta (arrowheads) move to the division site during ring constriction in WT, *ync13Δ*, and *rng10Δ* cells. Red boxes mark the ring position. Please also see the S1–S3 Videos. **(D)** Distribution of final trackable docking sites (dots and the box are colored the same for each cell) of Trs120 labeled puncta in 2-min movies during ring constriction. The color boxes mark the ring position as shown in **(C)**. The underlying data (for panels B and D) can be found in S1 Raw Data file. Bars, 5 μm.

which may be due to the delayed vesicle tethering (Fig 5C and 5D; S2 Video). Interestingly, in *ync13Δ* cells, more Trs120 concentrated within the division plane during and after ring constriction than in *rng10Δ* cells (Figs 5A, 5C, 5D and S8A; S3 Video). Consistently, in both *rng10Δ* and *ync13Δ* cells, it took longer for vesicles to fuse with the plasma membrane at division site compared to WT (Fig 5C). Thus, like Ync13, Rga7 and Rng10 also participate in the TRAPP-II complex-mediated exocytosis to recruit glucan synthases such as Bgs4 for septum formation during cytokinesis.

## Ync13 is a key functional effector of the Rga7-Rng10 complex for vesicle targeting during cell division

We found that Ync13, Rga7, and Rng10 form a complex that helps the TRAPP-II complex tether vesicles associated with secondary, but not primary, septum formation. To test whether Ync13 serves as a functional effector of the Rga7–Rng10 subcomplex, we mildly overexpressed Ync13 using the strong *3nmt1* promoter under repressing condition (YE5S rich medium with thiamine) and/or artificially retained Ync13 at the division site using the GBP-GFP system in *rga7Δ* cells. Specifically, *3nmt1-mECitrine-ync13* was retained at the division site by GBP-tagged Pmo25, which is an MO25 family protein that mostly colocalizes with Ync13 on the division plane, although Pmo25 also localizes to one of the spindle pole bodies [97–100]. Ync13 is not an abundant protein [58] and its division-site level is reduced by ~85% in *rga7Δ* cells (Fig 3B), thus mild overexpression is necessary for these assays. *rga7Δ* cells had severe lysis and negligible growth on plates with Phloxine B (a red dye accumulated in lysed or unhealthy cells because they cannot pump it out) at both 25 and 36°C (Fig 6A–6D). Mild overexpression of Ync13, but not expressing Pmo25-GBP alone, reduced cell lysis of *rga7Δ* cells from 78% to 51% at 36°C and promoted more growth at both 25 and 36°C (Fig 6A–6D, compare strains 1, 5, and 6). As expected, microscopy revealed robust localization of Ync13 at both the division site and the spindle pole body in *rga7Δ* cells expressing both *3nmt1-mECitrine-ync13 pmo25-GBP*, and this tethered Ync13 persisted along the cleavage furrow throughout cytokinesis and septum formation (Fig 6A and 6B, compare strains 3, 4, 5, and 7). Importantly, enforced division-site localization of Ync13 dramatically rescued the cytokinesis and cell lysis defects and growth of *rga7Δ* cells at both 25 and 36°C, as evidenced by further decreased cell lysis to 15% in *rga7Δ 3nmt1-mECitrine-ync13 pmo25-GBP* cells at 36°C (Fig 6A–6D; compare strains 1, 5–7). Together, these findings indicate that Ync13 is a key functional effector acting downstream of the Rga7–Rng10 complex at the division site.

To test if there is a correlation between Ync13 levels and vesicle fusion at the division site, we mildly overexpressed Ync13 from the *3nmt1* promoter in YE5S rich medium without additionally added thiamine to obtain cells with different Ync13 levels (the rich medium has some residual amount of thiamine, which partially represses the *nmt1* promoter) and then tracked the Rab11 GTPase Ypt3 labeled vesicles. This resulted in increased levels of Ync13 as well as Ypt3 at the division site (S8B Fig). We measured the Ync13 intensity at division site and counted the number of Ypt3 vesicles reaching the division site in 2-min continuous movie at the middle focal plane. We observed that increasing Ync13 level promoted the tethering and accumulation of more Ypt3 vesicles at the division site until it reached a plateau (S8B Fig). Collectively, Ync13, working with Rga7 and Rng10, plays an important role in vesicle targeting and fusion on the plasma membrane at the division site during cytokinesis.

## Electron microscopy reveals defective septum in *ync13Δ* cells

Because many more *ync13Δ* cells lyse when grown in YE5S rich medium than in the EMM5S minimal medium, we observed septum morphology after shifting cells from YE5S with sorbitol to YE5S by electron microscopy. After growing *ync13Δ* cells in YE5S for 3.5 h, >50% forming or closed septa were defective (Fig 7A–7C). This number was likely underestimated because many cells had lysed before the high-pressure freezing to prepare the samples for electron microscopy. Compared to *ync13+*, the septa in *ync13Δ* cells were distorted, curved, wavy, uneven, and/or thinner (Fig 7A and 7B), suggesting improper septum formation, especially the secondary septum. The septum in many cells was thin or missing at the edge of the division plane during daughter-cell separation, which led to cell lysis (Fig 7B, two cells at the bottom right). The primary septum had no obvious defects in *ync13Δ* cells (Fig 7B, arrowheads). We also observed some

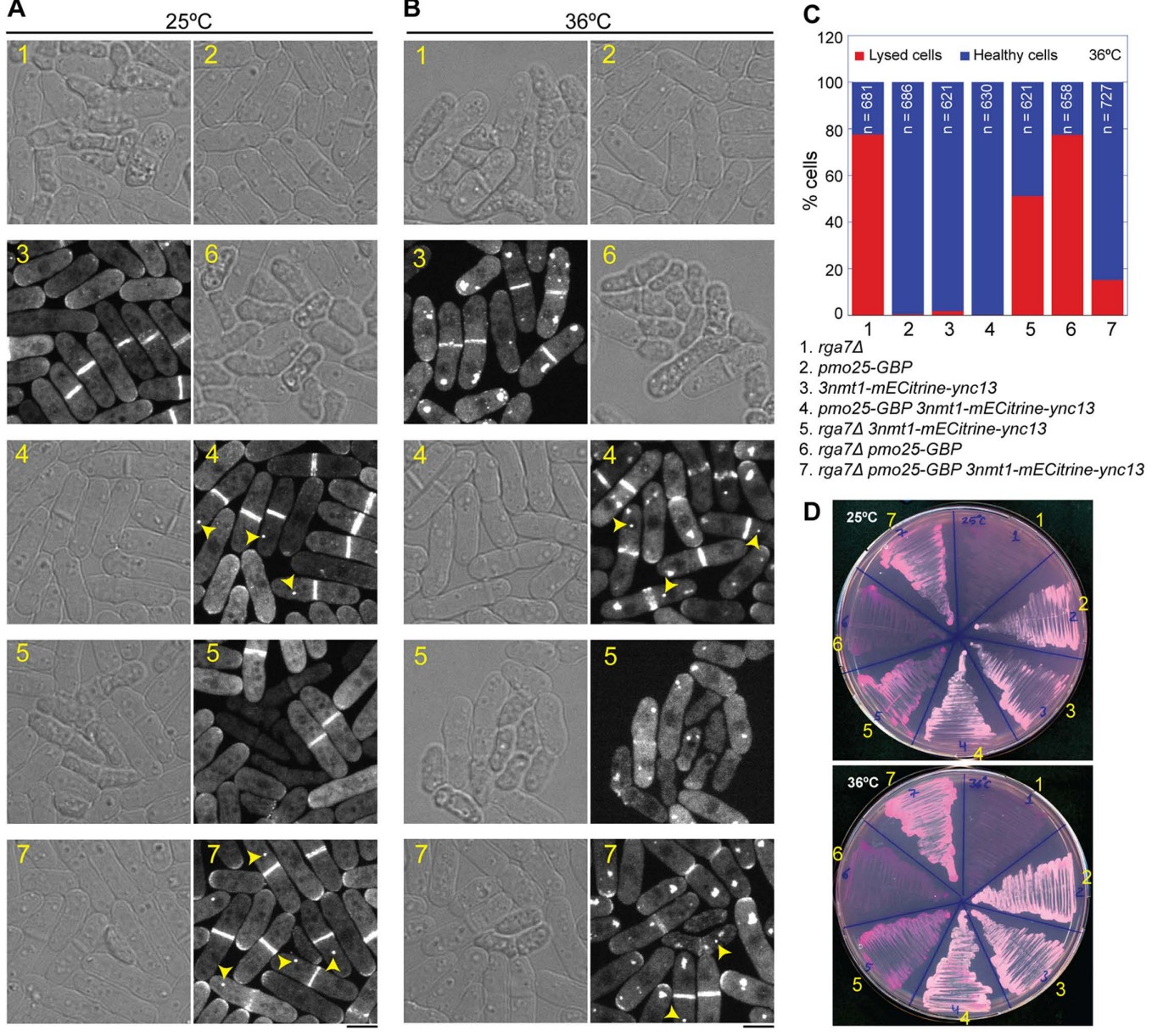

**Fig 6. Mild overexpression and forced retention of Ync13 at the division site rescue *rga7Δ*. (A–D)** The genotypes of the seven strains used are listed between panels **(C)** and **(D)**. **(A and B)** Micrographs showing cell morphology and/or Ync13 localization in the indicated strains in *rga7⁺* or *rga7Δ* cells at 25°C **(A)** or 36°C **(B)**. Cells were grown for 36–40 h at 25°C in YE5S + thiamine medium and then imaged directly (A) or shifted to 36°C for 3.5 h before imaging. Arrowheads mark the examples of mECitrine-Ync13 mistargeting to the SPB by Pmo25-GBP in strains 4 and 7. Note, mECitrine-Ync13 tends to form irregular aggregates at 36°C but not 25°C. Scale bars, 5 μm. **(C)** Quantification of cell lysis for cells grown at 36°C for 3.5 h as in **(B)**. **(D)** Growth assay illustrating the rescue of *rga7Δ* by tethering mECitrine-Ync13 to the division site using Pmo25-GBP or by mildly overexpressing Ync13 using the *3nmt1* promoter. Cells were grown on YE5S + PB plates for 24 h at 25°C or 36°C. The underlying data (for panel C) can be found in S1 Raw Data file.

electron-dense filamentous materials at the leading edge of the forming septum in some *ync13Δ* cells (arrow), which was rare in *ync13⁺* cells. These defects occured earlier during septum formation and were more severe than those described in published data when cells were grown in the EMM5S minimal media [58]. The electron microscopy results support

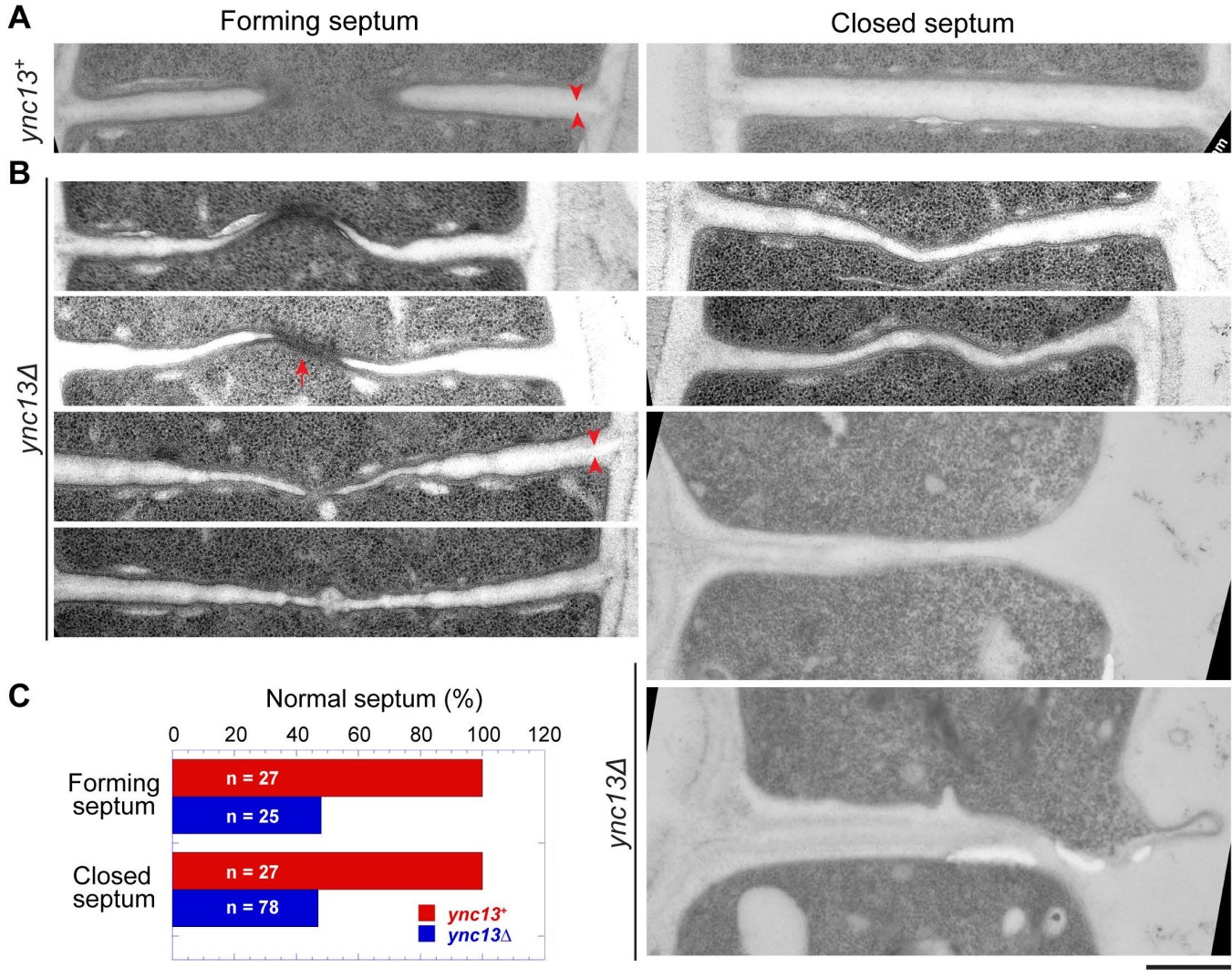

**Fig 7. The septum is defective in *ync13Δ* cells revealed by electron microscopy.** Electron microscopy thin sections were rotated and interpolated (bicubic) using Fiji so the septum is horizontal. Then the septum region was cropped and shown. Left: forming septum; Right: closed septum or daughter-cell separation and cell lysis (the two cells at bottom right). Cells were grown exponentially at 25°C in YE5S + 1 M sorbitol for ~48 h and then washed into YE5S without sorbitol and grown for 3.5 h before collection for high-pressure freezing. Arrowheads mark examples of the primary septum. Arrow marks an example of the electron-dense materials near the leading edge of the septum. **(A)** *ync13⁺* cells. **(B)** Representative *ync13Δ* cells with abnormal septa. **(C)** Quantification of percentage normal septa in *ync13⁺* and *ync13Δ* cells. The septa in **(B)** are defined as abnormal. The underlying data (for panel C) can be found in S1 Raw Data file and uncropped images in S2 Raw Images. Bar, 500 nm.

the idea that Ync13 is important for recruitment and maintenance of the glucan synthases Bgs4 and Ags1 on the plasma membrane for secondary septum formation during cytokinesis.

## Discussion

The interplay between membrane trafficking and septum formation or extracellular matrix remodeling is critical for maintaining cell integrity and successful cytokinesis from yeast to mammalian cells [101–107]. The (1,3)β-glucan synthase Bgs1/Cps1, recruited by Sbg1 and the contractile ring that is anchored by the F-BAR protein Cdc15 and likely other proteins (and lipids), is responsible for the primary septum formation [7–11,25,32,33]. Our findings elucidate the crucial roles of the

Ync13–Rga7–Rng10 complex in coordinating selective vesicle tethering and fusion mediated by the TRAPP-II complex and the SM protein Sec1 at the cleavage furrow (Fig 8). This module ensures precise and timely plasma-membrane deposition and secondary septum formation by the glucan synthases Bgs4 and Ags1 to prevent cell lysis during daughter-cell separation (Fig 8). While septins, the anillin Mid2, and the exocyst are mainly concentrated at the rim of the division plane to recruit the glucanases Eng1 and Agn1 to digest the primary septum during daughter-cell separation [34–42] (Fig 8).

## Rga7, Rng10, and Ync13 cooperatively regulate vesicle tethering and fusion during cytokinesis

Our findings have several mechanistic implications for how vesicle tethering and fusion are coupled during cytokinesis. First, the interaction between the Munc13 protein Ync13 and the SM protein Sec1 suggests that Ync13 functions as a SNARE-priming factor at the division site before SNARE complex assembly. Sec1 and its binding partner Mso1 bind

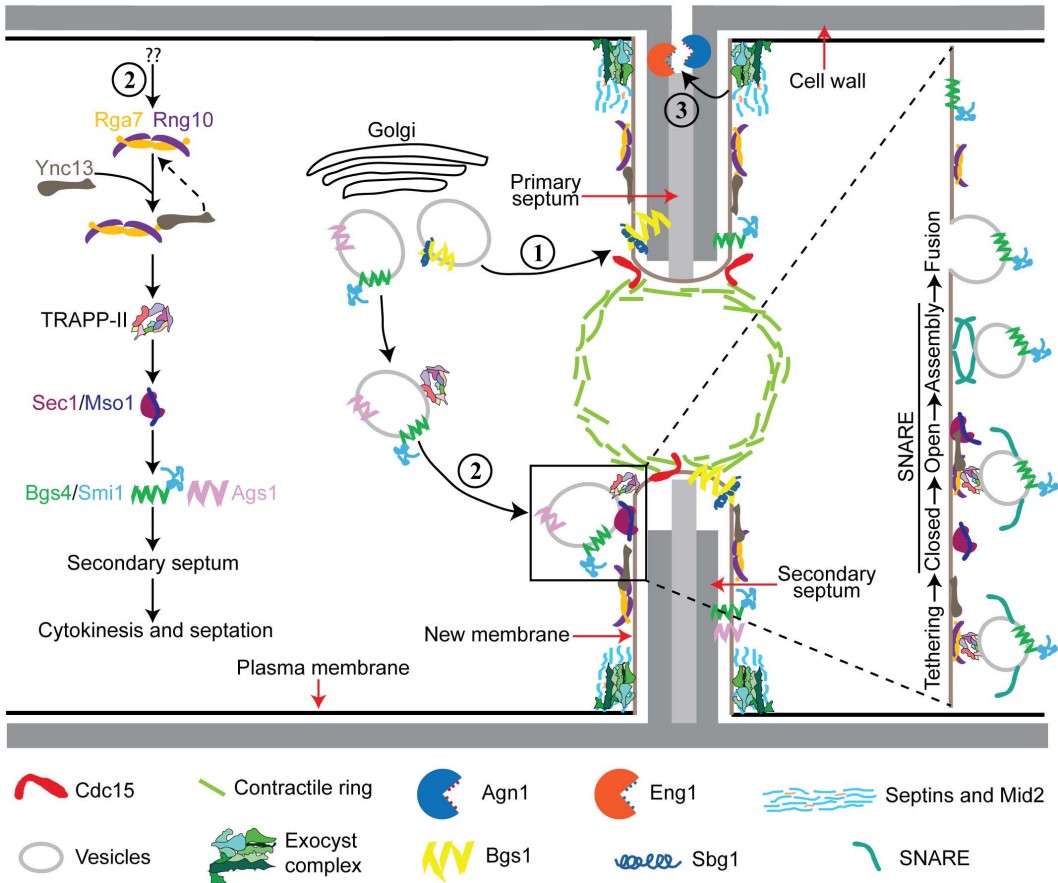

**Fig 8. Working model for the major pathways of plasma membrane deposition and septum formation during cytokinesis.** The septum is illustrated asymmetrically intentionally. The top half septum shows septum formation and digestion of the primary septum; the bottom half shows septum formation. (1) F-BAR protein Cdc15 (and others) anchor the contractile ring to the plasma membrane, which recruits the (1,3)β-glucan synthase Bgs1 and its adaptor Sbg1 for primary septum formation. (2) Association of the F-BAR protein Rga7 and coiled-coil protein Rng10 with the Munc13/UNC-13 protein Ync13 recruits the TRAPP-II complex along the cleavage furrow, which tethers vesicles and primes the SM protein Sec1 and its binding partner Mso1 for the SNARE complex assembly to trigger vesicle fusion with the plasma membrane. This pathway maintains normal levels and distribution of gluacan synthases Bgs4/Smi1 and Ags1 for secondary septum formation. (3) The anillin Mid2 and septin rings restrict most of the exocyst complex to the rim of cleavage furrow to tether secretory vesicles loaded with the glucanases Eng1 and Agn1 and other cargos for digestion of primary septum to trigger daughter-cell separation.

syntaxin-family SNAREs and are essential for polarized secretion [57,108–110]. In other systems, SM proteins work in tandem with Munc13 proteins to assemble fusogenic SNARE complexes [54,55,90,92]. Consistently, we found that mistargeted Ync13 is able to recruit Sec1 to mitochondria, even without the coiled-coil protein Rng10 (S2G Fig). Moreover, Ync13 overexpression enhanced tethering and/or accumulation of Ypt3-labeled vesicles at the division site (S8B Fig), further supporting a direct role of Ync13 in promoting vesicle docking and fusion, analogous to the established functions of mammalian Munc13 proteins [53,54,90,92], which need to be tested by in vitro reconstitution assays in future studies. These findings suggest that Ync13 has the capacity to act independently of Rng10 in driving vesicle fusion, although it normally operates in collaboration with the Rga7–Rng10 complex.

Second, Ync13, by binding to the lipid-binding protein complex Rga7–Rng10 [74,75], is well positioned to help tether or recruit vesicles to the division site, where its interaction with Sec1 could facilitate the transition from a tethered vesicle to a docked, SNARE-engaged vesicle ready for fusion. Third, Ync13's direct interaction with the Rga7–Rng10 complex links Ync13 indirectly to the TRAPP-II vesicle tethering complex. With a $K_d$ of 10 µm, the interaction of between Ync13 and Rga7 is not very strong, which is consistent with their dynamic bindings to the plasma membrane revealed by FRAP assays (half times, 2–3 s for Ync13 and 23 s for Rga7) [58,74]. The dynamic and weak interaction is consistent with their roles in highly dynamic processes of vesicle fusion. We observed that Rga7 and Rng10 selectively interact with the TRAPP-II but not with the exocyst complex in mislocalization experiments. The absence of interaction of Ync13, Rng10, or Rga7 with the exocyst complex indicates that Ync13 performs a TRAPP-II-specific function on vesicle tethering. This is reinforced by the distinct accumulation of Trs120 puncta in rng10Δ and ync13Δ mutants. In rng10Δ cells, TRAPP-II-marked vesicles often fail to arrive at the cleavage furrow and accumulate outside the division site. In contrast, in ync13Δ cells, vesicles arrive at the division furrow but linger there longer before fusion. We speculate that, similar to Munc13 [56,88,111], the function of Ync13 may be attributed to a post-vesicle docking stage after tethering. When the TRAPP-II-mediated tethering is compromised by rga7Δ and rng10Δ, the vesicles cannot efficiently contact with its target membrane surface and move outside of the division site. However, in ync13Δ cells, the physical interaction between vesicles and the target membrane may have been established after tethering process, but the docking process is affected, which leads to the delayed vesicle fusion. Roles of the TRAPP-II complex in cytokinesis have been reported in Drosophila and plant cells [112,113]. Taken together, we propose that the Rga7–Rng10 complex serves as a landmark for vesicle tethering by the TRAPP-II complex and interacts with Ync13 to facilitate the assembly of the SNARE complex and promote membrane fusion.

The Pmo25-GBP mislocalization and Ync13 mild overexpression experiments provide direct evidence that Ync13 is a key functional effector acting downstream of the Rga7–Rng10 complex (Fig 8). By artificially tethering Ync13 to the division site through Pmo25, we were able to partially bypass the requirement for Rga7 and Rng10 in recruiting Ync13. Forced localization and overexpression of Ync13 at the division site not only rescue the cytokinetic defects and cell lysis of rga7Δ mutant but also restore cell growth (Fig 6). These findings demonstrate that the essential role of the Rga7–Rng10 complex is to position Ync13 at the division site, where it can execute its effector function. However, Ync13 also feedbacks onto Rga7 and Rng10, because their localization pattern at the division site depends on Ync13 (Fig 8).

Munc13 functions in neurotransmitter release by priming vesicle tethering process for sudden release of synaptic vesicle pool [114]. This ensures a reliable and speedy neurotransmission following synaptic collapse. Ync13 regulates both exocytosis and endocytosis. However, the exact mechanisms and its interactions with other proteins were unknown before this study. Our previous studies show that Rga7 binds the C-terminal portion of Rng10 to increase Rga7's avidity with the plasma membrane [74,75]. Our AlphaFold3 modeling provides a structural framework for understanding how Ync13, Rga7, and Rng10 form a functional module at the division site. The predictions suggest that Rga7 uses its F-BAR domain to engage Rng10 while its GAP domain to interact with Ync13, thereby acting as a bridge that links the lipid-binding protein Rng10 to the vesicle-fusion factor Ync13 (S5 Fig). The predicted interaction has to be tested in future experiments. This hierarchical organization explains our experimental findings that Ync13 does not mislocalize to mitochondria with Rng10(751-1038) in rga7Δ cells and that Rga7 is strictly required for assembling the ternary complex. The model also

reinforces the idea that Rga7 couples membrane curvature sensing (via the F-BAR domain) with vesicle docking and fusion (via the GAP domain–Ync13 interaction), positioning Ync13 at the cleavage furrow where it can promote Sec1 recruitment and SNARE engagement. Together, the AlphaFold3 predictions integrate well with our mislocalization and genetic data, providing a mechanistic explanation for how the Rga7–Rng10–Ync13 module coordinates vesicle tethering and fusion during cytokinesis. Together, our studies here indicate that Rga7–Rng10 interacts with Ync13, which interacts with the SM protein complex Sec1/Mso1. These proteins play a crucial role for tethering the secretory vesicles by regulating the TRAPP-II and SNARE complexes, which are essential for fusion of vesicles with target membranes (Fig 8).

## Selective recruitment of the glucan synthases for the secondary septum by the Rga7–Rng10–Ync13 module at the division site

The transmembrane glucan synthases Bgs1, Bgs4, and Ags1 are essential for septum formation during cytokinesis and are maintained at the proper levels on the plasma membrane via membrane trafficking [7,9–12,15,33,115]. The primary septum in fission yeast is composed of mainly linear (1,3)β-glucan chains synthesized by Bgs1 that plays similar role as chitin in budding yeast [8]. Bgs1 concentrates behind the contractile ring, while Ags1 and Bgs4 localize at the cleavage furrow on both layers of the new plasma membrane to construct secondary septum. At low restrictive temperatures, the lethality of mutant sid2, the most downstream kinase in the Septation Initiation Network, is partially rescued by upregulating Rho1. Thus, it has been suggested that the Septation Initiation Network activates Rho1, which in turn activates the glucan synthases [116].

Our data demonstrate that the Rng10–Rga7–Ync13 module specifically recruits Bgs4 and Ags1 for secondary septum assembly, but not Bgs1 for primary septum synthesis. Ectopic targeting of Rga7 or Rng10 is sufficient to redirect vesicles containing Bgs4 and Ags1 to mitochondria. However, Bgs1 is not mislocalized under these conditions. When overexpressed, Ync13 can also pull Bgs4 and Ags1 to mitochondria, presumably through Ync13's binding to Rga7/Rng10. Importantly, the proper spatial distribution of Bgs4 and Ags1 relies on the functional interaction between Ync13, Rga7, and Rng10. In WT cells, Rga7, Rng10, Ync13, Bgs4, and Ags1 colocalize at the division plane and are mostly evenly distributed around the edges of the invaginating septum and throughout the maturing septum. In contrast, in ync13Δ cells, all these proteins abnormally concentrate at the leading edge of the furrow and later at the center of the division plane, forming a flattened convex structure. This mislocalization depends on Rga7/Rng10 and leads to a significant reduction in Bgs4 (and to a lesser extent Ags1) near the septum periphery, highlighting the critical role of Ync13 in maintaining proper enzyme distribution. The lack of proper Bgs4 deposition on the plasma membrane at the septum edges likely compromises the thickness and rigidity of the new cell wall, explaining why ync13Δ, rga7Δ, and rng10Δ mutants all undergo cell lysis during cell separation. In contrast, the primary septum synthase Bgs1 remains correctly localized even in ync13Δ cells, suggesting that primary septum formation follows a Ync13/Rga7/Rng10-independent pathway. Thus, our study highlights the distinct roles of Rga7–Rng10–Ync13 module in regulating the formation of the secondary septum and maintains cell wall integrity by recruiting the secondary septum glucan synthases Bgs4 and Ags1 at the division site.

Our findings have two important mechanistic implications. First, the fact that the Rng10–Rga7–Ync13 complex only recruits Bgs4 and Ags1, but not Bgs1, strongly indicates that there are at least two distinct types of secretory vesicles to transport cell wall synthases. In the budding yeast S. cerevisiae, different types of secretory vesicles have been reported [117,118]. The constitutive or low-density secretory vesicles deliver plasma membrane proteins, cell wall components, and enzymes necessary for cell growth from trans-Golgi network. Invertase-containing or high-density secretory vesicles transport stress-induced cargos, such as the enzyme invertase [119,120]. Second, the central and leading edge localization of Rga7, Rng10, Bgs4, and Ags1 in ync13 mutants suggests that the Bgs4 and/or Ags1-containing vesicles are mis-directed to where the Bgs1 vesicles are normally targeted, perhaps using the machinery that the Bgs1 vesicles utilize. It will be interesting to test these possibilities in future studies.

Rho GTPases are small molecular switches that regulate multiple cellular processes including cytokinesis [121]. Of the six Rho GTPases (Rho1-5 and Cdc42) in fission yeast, Rho1 and Rho2 play crucial roles in maintaining cell integrity during cytokinesis [14,122–125]. Activated mainly by Rho GEF Rgf3 and its adapter arrestin Art1, GTP-bound Rho1 activates β-glucan synthases Bgs1 and Bgs4 and protein kinase Cs Pck1 and Pck2 [27,124–130]. Similarly, Rho2 functions in the cell integrity pathway and activates α-glucan synthase Ags1 for septum formation [14,125]. However, it is unclear how Rho GTPases regulate recruitment, distribution, and maintenance of glucan synthases.

More studies are available in budding yeast regarding the regulations of septum or cell wall synthases. The primary septum in the budding yeast *S. cerevisiae* consists of the chitin synthesized by chitin synthases Chs1 and Chs2, which functions similarly to the primary septum of fission yeast consisting of linear (1,3)β-glucan [8,131–134]. Cdc14 dephosphorylates Chs2 for its localization at the septation site from ER [133,135]. Chs3 is responsible for the formation of a chitin ring at the emerging bud and of the chitin dispersed in the cell wall. Chitin synthase III complex also contains Chs4 that anchors to the plasma membrane and interacts with Chs3 and Bni4 [136]. Septins interact with Bni4 to hold this complex together which help the chitin synthase to synthesize sufficient chitins for the primary septa. Chitin synthases localize in Rho1-dependent manner [137]. The secondary septum in budding yeast is made up of β-glucans with minor amount of chitin. Rho1 and Rho2 activate FKS1 by interacting with the glycosyltransferase domain and the transmembrane helix [138–141]. These interactions induce conformational changes to push growing (1,3)β-glucan chain [142]. Alternatively, secondary septum synthesis can also be achieved by transcriptional activation of *FKS2* and *GFA1* genes by the cell wall integrity pathway [21]. Cdc42 negatively regulates secondary septum formation [141,143–145]. However, it is still poorly understood how the glucan synthases FKS1/2 are recruited to the division site in budding yeast.

By identifying the Rga7–Rng10–Ync13 module as the main pathway for proper recruitment and distribution of the glucan synthases Bgs4 and Ags1 at the division site for secondary septum formation, our current study provides significant insights into the regulatory mechanisms of glucan synthases, which are ideal targets for antifungal drugs. However, cells must possess other unidentified mechanisms for Bgs4 and Ags1 recruitment because *ync13Δ* or *rga7Δ rng10Δ* cells can survive in medium with sorbitol or in minimal medium, although not in rich medium. Rga7 is Rho2 GAP [76,146,147], thus it is possible that the Rga7–Rng10–Ync13 module regulates Bgs4 and Ags1 via Rho GTPases. However, the functions of Rga7's GAP domain remain poorly understood [73,75,76]. Moreover, no connection between FKS1/2 and the homologs (RGD1 and YOR296W, respectively) of Rga7 and Ync13 in budding yeast have been established.

In summary, in this study, we find that Ync13, Rga7, and Rng10 form a regulatory module that links vesicle trafficking with the precise targeting and distribution of glucan synthases Bgs4 and Ags1 to ensure efficient septum formation. Ync13 plays a key role in both vesicle fusion and spatial distribution of Rga7/Rng10, underscoring its crucial function in cytokinesis. Moreover, the selective and dynamic interactions between Rga7/Rng10, TRAPP-II, and glucan synthases fine-tune vesicle targeting specificity. These findings deepen our understanding of how membrane trafficking processes are spatially and temporally coordinated to enable successful cytokinesis.

## Materials and methods

### Strain construction and growth methods

The strains used in this study are listed in S2 Table. We constructed the strains by standard crosses of cells with opposite mating type or tagging genes using the PCR and homologous recombination-based gene targeting method so fusion genes were expressed at endogenous chromosomal loci and under the control of native promoters except where noted [148,149]. The exceptions are tagged glucan synthases Bgs1, Bgs4, and Ags1, which are regulated by their native promoters but integrated at the *leu1* loci with the endogenous copies deleted [7,9,10,12,150,151]; and the *3nmt1-mECitrine-ync13* strains, where the *nmt1* promoter is regulated by thiamine. Cells were grown at exponential phase ($OD_{595} < 0.5$) at

25°C in rich YE5S (yeast extract with five supplements) liquid medium for ~48 h before imaging (more details below) or other experiments except where noted.

To tag Ync13 with 3xFLAG for mass spectrometry, the fragment of GGSGGS-3xFLAG was first inserted at the SmaI site in pREP3x plasmid vector. FL Ync13 cDNA was cloned into pREP3x-GGSGGS-3xFLAG between the *3nmt1* promoter and GGSGGS-3xFLAG tag by Gibson assembly [152]. The plasmids pREP3x-Ync13-FL-3xFLAG and pREP3x-3xFLAG (control) were then transformed into protease-deficient strain TP150 (*leu1-32 SM902*) [153,154] and positive transformants were selected on EMM5S-leucine minimal medium. The positive colonies were grown on EMM5S-leucine liquid medium to $OD_{595} = 1.0$ before collection and lyophilization.

### Mass spectrometry

We used the previously described protocol for purification of S-tagged proteins with several modifications to identify the Ync13 binding partners [155]. Approximately 2 g of lyophilized cells (TP150 strains with plasmid pREP3x-Ync13-FL-3xFLAG or pREP3x-3xFLAG control) were broken by grinding with a mortar and pestle at room temperature until the cells became a homogenous fine powder. For protein extraction, the cell powders were thoroughly mixed with 25 ml of cold HK extraction buffer (25 mM Tris, pH 7.5, 0.5% NP-40, 300 mM NaCl, 5 mM EDTA, 15 mM EGTA, 60 mM β-Glycerophosphate, 500 μM $Na_3VO_4$, 10 mM NaF, 1 mM PMSF, 1 mM DTT, and protease inhibitors [Roche]). The cell extracts were cleared by two rounds of centrifugation at 4°C (21,000 rpm for 30 min, 21,000 rpm for 10 min). Then, cell extracts were incubated with 200 μL anti-FLAG M2 affinity gel (A2220, Millipore) at 4°C for 2 h. The beads were collected by centrifugation at 4,000 rpm and washed once with an equal volume of HK extraction buffer, 4× with an equal amount of washing buffer (25 mM Tris, pH 7.5, 300 mM NaCl, 5 mM EDTA, 500 μM $Na_3VO_4$, 10 mM NaF, 1 mM PMSF, and 1 mM DTT), and 2× with 1 ml of washing buffer. The proteins on the beads were eluted by incubating with 500 μl of 200 μg/ml 3xFLAG peptide (F4799, Sigma-Aldrich) at 4°C for 30 min. For mass spectrometry analysis, the samples were run through the SDS–PAGE gel. Protein bands except the Ync13 band were excised as one sample and processed for mass spectrometry (Mass Spectrometry and Proteomics Facility, The Ohio State University).

### Confocal microscopy and image analysis

Cells from −80°C stocks were streaked onto YE5S plates and grown at 25°C for about 2 d and then fresh cells were inoculated into YE5S liquid medium and grown in the log phase for ~48 h at 25°C before imaging except where noted. For strains requiring osmotic stabilizer to survive, such as strains with *ync13Δ, ync13-19 rng10Δ*, *rng10Δ rga7Δ*, and some strains with Tom20-GBP, cells were woken up on YE5S + 1.2 M sorbitol plate and grown in YE5S + 1.2 M sorbitol liquid medium at log phase for 36–48 h, and then were washed with YE5S without sorbitol and grown in YE5S for 4 h before imaging except where noted. For comparison, some strains with *ync13Δ, ync13-19, rga7Δ*, and *rng10Δ*, and their double mutants of certain combinations were grown in EMM5S for 48 h before imaging. These strains have less severe phenotype or cell lysis in EMM5S than in YE5S-rich medium. For Pmo25-GBP mistargeting and rescue experiments, cells were grown for 36 h at 25°C in YE5S + thiamine (final concentration 5 μg/ml) and imaged on YE5S + thiamine gelatin pads directly, or shifed to 36°C for 3.5 h and then imaged at 25°C. To test the tethering and fusion of tdTomato-Ypt3 labeled vesicles at different Ync13 concentrations, *3nmt1-mECitrine-ync13* cells were grown in YE5S medium with thiamine for 24 h and then overexpressed in YE5S medium without thiamine for 16 h before imaging.

Confocal microscopy was done as previously described [13,35,57,129]. Briefly, cells were collected by centrifugation at 3,000 rpm for 30 s and washed once with 1 ml EMM5S and once 1 ml EMM5S containing n-propyl gallate at a final concentration of 5 μM to reduce autofluorescence and protect cells from free radicals during imaging [80,156]. We imaged cells on glass slides with a gelatin pad (20% gelatin in EMM5S + 5 μM n-propyl gallate) at ~23°C. For long movies, cells were washed with 1 ml EMM5S + 5 μM n-propyl gallate and placed onto a coverglass-bottom dish (Delta TPG Dish; Biotechs, Butler, PA, United States) and then covered with a piece of EMM5S agar [157]. For fluorescence microscopy at 36°C, the cells were grown at

25°C for ~2 d and then shifted to 36°C and grown for a given time (see figure legends). Before imaging, cells were washed and concentrated in pre-warmed YE5S liquid medium with 5 µM n-PG. Then 10 µl of the concentrated cells were spotted onto a coverglass-bottom dish, covered with the prewarmed YE5S agar, and imaged at 36°C in a preheated climate chamber (stage top incubator INUB-PPZI2-F1 equipped with UNIV2-D35 dish holder; Tokai Hit, Shizuoka-ken, Japan).

For most fluorescence images and time-lapse movies, cells were imaged using a spinning-disk confocal system (Ultra-VIEW Vox CSUX1 system; PerkinElmer, Waltham, MA) with 440-, 488-, 515-, and 561-nm solid-state lasers and a back-thinned electron-multiplying charge-coupled device (EMCCD) camera (C9100-23B; Hamamatsu Photonics, Bridgewater, NJ) on a Nikon Ti-E microscope without binning [23,42]. For Figs 1F, 6, S2G, S4H, S7, and S8B, cells were imaged using a Nikon spinning-disk confocal system (W1 + SoRa) with 488 and 561 nm solid-state lasers and an ORCA-Quest qCMOS camera (C15550; Hamamatsu Photonics, Bridgewater, NJ) on a Nikon Eclipse Ti-2E microscope with 2 × 2 binning [158].

Images were analyzed using Volocity (PerkinElmer), NIS-Elements (Nikon), and ImageJ/Fiji software (National Institutes of Health, Bethesda, MD). Fluorescence images shown are single middle focal plane or maximum-intensity projections of image stacks with 0.5 µm spacing except where noted. To measure protein levels across the division plane, the cells after constriction of the contractile ring (Rlc1 ring constricted to a spot at cell center) except where noted were chosen and rotated so that the septa were horizontal. A 23 × 6 pixel region of interest (ROI) was drawn to cover the protein signal at division site. The plot profile of the ROIs was recorded.

We tracked secretory vesicles similarly as before [6,58]. Briefly, the middle focal plane of cells was imaged with a speed of 2–5 frames per second (fps) for the channel with fluorescently labeled vesicles in 2-min movies. The Rlc1 channel was imaged once every minute. DIC images were taken as snapshots immediately before and after the fluorescence movies to make sure no focal shifting. The movements of vesicles were tracked manually using ImageJ plug-in mTrackJ [159]. The data coordinates were then transformed by Matlab software so that the septa were horizontal. The cell width was normalized to 4 µm before plotting. Statistical analyses were performed using Welch's or two-tailed Student $t$ test in this study.

## Plasmid construction, protein purification, in vitro binding assays

The fragment of MBP-TEV-GGSGGS was first cloned into pET21a vector before BamHI site by Gibson assembly to construct pET21a-MBP vector [152]. Ync13 FL cDNA was cloned into pET21a-MBP vector between the GGSGGS linker and the 6His tag by Gibson assembly. FL Rga7 was cloned into the pET21a vector between the T7 tag and the 6His tag by Gibson assembly. The constructs were confirmed by sequencing.

We purified recombinant proteins by transforming the plasmids into BL21 (DE3) pLysS cells (Novagen). MBP-Ync13-6xHis expression was induced with 0.2 mM IPTG at 17°C for 36–48 h. Rga7-6xHis was expressed with 0.5 mM IPTG at 25°C for 15 h. Purifications of 6His-tagged proteins were carried out as previously described [160,161]. Briefly, the proteins were purified with Talon metal affinity resin (635501; Clontech, Mountain View, CA) in extraction buffer (50 mM sodium phosphate, pH 8.0, 400 mM NaCl, 10 mM $\beta$-mercaptoethanol, 1 mM PMSF, and 20 mM imidazole) with EDTA-free protease inhibitor tablet (Roche) and eluted with elution buffer (50 mM sodium phosphate, pH 8.0, 400 mM NaCl, 10 mM $\beta$-mercaptoethanol, 1 mM PMSF, and 300 mM imidazole). The purified proteins were then dialyzed into the binding buffer (137 mM NaCl, 2 mM KCl, 10 mM $Na_2HPO_4$, 2 mM $KH_2PO_4$, 0.5 mM dithiothreitol, and 10% glycerol, pH 7.4).

For in vitro binding assays between MBP-Ync13-6xHis and Rga7-6xHis, purified proteins were dialyzed into the binding buffer. We incubated 1 ml MBP-Ync13-6xHis (2 µM) or 75 µl MBP-6xHis (27 µM) control with 500 µl Amylose beads for 1 h at 4°C and washed the beads 8× with 1 ml of the binding buffer each time to remove unbound proteins. Then 1 ml Rga7-6xHis (10 µM) was incubated with the 100 µl beads with MBP-Ync13-6xHis or MBP-6xHis for 1 h at 4°C. After washing 4× with 1 ml of the binding buffer each time, the beads were boiled with sample buffer for 5 min. Then the samples were run on SDS–PAGE gel and detected with Coomassie Blue staining. For measuring the $K_d$ between MBP-Ync13-6His and Rga7-6xHis, we followed the methods and guidelines as described [75,162,163]. Rga7-6xHis at 4 µM was titrated with

MBP-Ync13-6xHis immobilized on Cobalt beads or the same volumes of beads with MBP-6xHis. Beads were pelleted at 16,000$g$, and proteins in supernatant were separated by SDS–PAGE, stained with Coomassie, and scanned to measure and calculate the fractions of proteins bound to the beads.

### Co-IP and Western blotting

Co-IP and Western blotting were performed as described except where noted [42,80,164,165]. Briefly, proteins tagged with mEGFP, mECitrine, GFP, or 13Myc were expressed under their native promoters. Lyophilized cells were ground into a homogeneous fine powder using pestles and mortars. IP buffer (50 mM 4-(2-hydroxyethyl)-1-piperazineethanesulfonic acid [HEPES], pH 7.5, 150 mM NaCl, 1 mM EDTA, 0.1% NP-40, 50 mM NaF, 20 mM glycerophosphate, and 0.1 mM $Na_3VO_4$, 1 mM PMSF, and protease inhibitor [Roche] 1 tablet/30 ml buffer) was added at the ratio of 10 µl: 1 mg lyophilized cell powder. Sixty µl Dynabeads protein G beads (Invitrogen) were incubated with 5 µg polyclonal GFP antibody (Novus Bio) for 1 h at room temperature. After three washes with PBS and one wash with 1 ml IP buffer, the beads were incubated with cell lysate for 2 h at 4°C. After 5 washes at 4°C with 1 ml IP buffer each time, the beads were boiled with 80 µl sample buffer. The protein samples were separated with SDS–PAGE gel and detected with monoclonal anti-GFP antibody (1:1,000 dilution; 11814460001; Roche, Mannheim, Germany), or monoclonal anti-Myc antibody (1:500 dilution, 9E10, Santa Cruz Biotechnology, Dallas, TX). Tubulin detected by the TAT1 antibody was used as a loading control [42]. Secondary anti-mouse immunoglobulin G (1:5,000 dilution; A4416, Sigma-Aldrich) was detected using SuperSignal Maximum Sensitivity Substrate (Thermo Fisher Scientific) on iBright CL1500 imager (Thermo Fisher Scientific) or other imagers.

### Alphafold prediction of the Ync13–Rga7–Rng10 complex

To gain mechanistic insights into the organization of the Ync13–Rga7–Rng10 module, we performed structural predictions using AlphaFold3 [166]. Previous work demonstrated that the F-BAR protein Rga7 forms a homodimer and that its F-BAR domain binds to the C-terminal (aa751-1038) region of Rng10 [74,75]. Based on these findings, we constructed a modeling system containing two FL Rga7 subunits, two Rng10(751–1038) fragments, and one FL Ync13 protein. For domain-specific predictions, truncated amino acid sequences were used as input. Sequences were retrieved from PomBase, and input files were prepared in FASTA format. Predictions were carried out using the AlphaFold3 multimer for heteromeric protein–protein interactions. Five models were generated and structural confidence was assessed using pLDDT and predicted aligned error scores. Structural visualization and analysis were performed in PyMOL (Schrödinger, LLC).

### Electron microscopy

We grew $ync13\Delta$ and $ync13^+$ cells exponentially at 25°C in YE5S + 1 M sorbitol for ~48 h and collected cells by spinning at 2,200 rpm for 3 min. Cells were then washed twice with equal volume of YE5S medium without sorbitol and diluted to the appropriate density and grown 3.5 h at 25°C. The $OD_{600}$ of cells was <0.5 before collecting for high-pressure freezing. Sample preparations and electron microscopy were performed as described previously [167,168]. Briefly, concentrated cell slurry (~2 µl) was transferred onto specimen carriers (Wohlwend type A, 3 mm wide, 0.1 mm deep) and covered with a flat lid (Wohlwend type B). The carrier sandwich was immediately processed by high-pressure freezing on a Wohlwend HPF Compact 02. The frozen samples were stored in liquid nitrogen. We opened the carrier sandwich in liquid nitrogen before freeze substitution, which used 1% uranyl acetate in acetone and embedded in Lowicryl HM20 on the Leica AFS 2 robot. Thin sections of 60 nm were cut with a diamond knife using a Leica Ultracut UC7 ultramicrotome and loaded onto carbon-coated 200-mesh copper grids (AGS160; Agar Scientific). The grids were post-stained with 2% uranyl acetate, and then Reynolds lead citrate for 10 min. We observed and imaged the grids using a FEI Tecnai 12 at 120 kV with a bottom-mount FEI Eagle camera (4k × 4k).

PLOS Biology

## Supporting information

**S1 Fig. Representative controls for the Tom20-GBP mistargeting experiments. (A–I)** Micrographs of DIC, 514/488/561 nm channels, and merged channels showing cells expressing Tom20-GBP and another indicated protein. Tom20-GBP does not bind to mCherry, RFP, or tdTomato so the tagged proteins cannot be recruited to mitochondria without proteins tagged with mEGFP or mECitrine. No signal bleed-through between red (561 nm)/yellow (514 nm) or red (561 nm)/green (488 nm) channels was detected. The brightness and contrast were adjusted the same as the experimental groups with three tagged proteins shown in Figs 1, S2, or S3 so some panels appear almost totally black. Bars, 5 μm.
(TIF)

**S2 Fig. Positive and negative physical interactions revealed by mistargeting to mitochondria using Tom20-GBP.** Mislocalized Ync13 ectopically targets (examples marked with arrowheads) Rga7 and Rng10 **(A)**, and Sec1 **(B)** to mitochondria; but cannot interact with the exocyst subunit Sec3 **(C)** or endocytic proteins Ede1 **(D)**, fimbrin Fim1 **(E)**, or clathrin light chain Clc1 **(F)**. Ync13 was mildly overexpressed using the *3nmt1* promoter by growing exponentially in YE5S liquid medium for ~48 h before imaging. **(G)** Mislocalized Ync13 ectopically targets Sec1 to mitochondria without Rng10 (examples marked with arrowheads). Cells were grown in YE5S + 1.2 M sorbitol + thiamine at 25°C for 24 h and then shifted to the same medium without thiamine to induce Ync13 expression for 16 h. Cells were washed twice with EMM5S and imaged on EMM5S gelatin pad. Bars, 5 μm.
(TIF)

**S3 Fig. Rga7 and Rng10 cannot mistarget Sec3 or Bgs1 to mitochondria and mislocalized Ync13 ectopically targets Bgs4 and Ags1 to mitochondria. (A and C)** Rga7 and Rng10 cannot mistarget the exocyst subunit Sec3 **(A)** or Bgs1 **(C)** to mitochondria. **(B)** Mislocalized Ync13 ectopically targets Bgs4 and Ags1 to mitochondria (examples marked with arrowheads). Ync13 was overexpressed using the *3nmt1* promoter by growing exponentially in YE5S liquid medium for ~48 h before imaging. Bars, 5 μm.
(TIF)

**S4 Fig. Physical interactions between full-length Ync13 and Rga7 or Rng10 truncations revealed by ectopically mistargeting to mitochondria by Tom20-GBP. (A)** Domain schematics of Rga7 and Rng10 [74,75]. **(B–G)** Arrowheads mark examples of colocalization at mitochondria. Except Rng10-(1-200) in (D), all other Rga7 and Rng10 truncations (B, C, and E–G) can mistarget Ync13-tdToamto to mitochondria. Rga7FBD = Rga7(1-320) [76]; Rga7(ΔF-BAR), Rga7 without the F-BAR domain. **(H)** mECitrine-Rng10(751-1038) cannot mistarget Ync13 to mitochondria without Rga7. Cells were grown in YE5S + 1.2 M sorbitol at 25°C for 36 h and then washed twice with EMM5S and imaged on EMM5S gelatin pad. Bars, 5 μm.
(TIF)

**S5 Fig. AlphaFold3 modeling of the Rga7–Rng10–Ync13 complex. (A)** Ribbon diagram of the predicted Rga7–Rng10–Ync13 complex, illustrating the overall domain organization and relative orientation of the three proteins. For the modeling, one copy of FL Ync13 (magenta), the dimer of FL Rga7 (green), and two copies of Rng10(aa751-1038) (orange) were used. The contacts between residues with a distance <4 Å are colored in cyan in (A–E). **(B)** Overlay (Root Mean Square Deviation: RMSD = 0.761 Å) of predicted structures of Ync13 N-terminus (aa 1-600) and Rga7-GAP (aa 321-695) domain panel (D) onto the whole complex in panel (A). Here, Ync13-N is colored in blue and Rga7(GAP) in yellow from (D) while all the original colors for the whole complex from (A). **(C–E)** Ribbon diagram of predicted interaction between (C) Rga7 F-BAR dimer (pTM: 0.74, ipTM: 0.71); (D) Ync13 N-terminal (1-600 aa) and Rga7-GAP domain (321-695 aa) (pTM: 0.63, ipTM: 0.64); and (E) Rga7 F-BAR and Rng10(751-1038) (pTM: 0.56, ipTM: 0.78). **(F)** The PAE (Predicted Aligned Error) plot for the AlphaFold3 predictions in (C–E).
(TIF)

**S6 Fig. Quantification of proteins from Western blots and localizations of Bgs4, Rga7, and Rng10 in temperature-sensitive mutants. (A and B)** Protein levels of Ync13 and Rga7 from the Western blots associated with Fig 3C (A) and Fig 3D (B). **(C)** Western blot and quantification of Rng10 levels in the indicated strains. **(D–G)** Cells grown exponentially at 25°C were shifted to 36°C for 4 h (D, F, G) or 2 h (E) before imaging. Rlc1-tdTomato as the ring marker. Bgs4 localization in *sec1-M2* (D) or *trs120-ts1* (E) mutant cells. Rga7 (F) and Rng10 (G) localization in *bgs4* mutants *cwg1-1* and *cwg1-2*. The underlying data (for panels A–C) can be found in S1 Raw Data file and uncropped Western blots (for panel C) in S1 Raw Images. Bars, 5 µm.
(TIF)

**S7 Fig. The spatial distribution of the β-glucan synthase Bgs1 at the division site is not significantly affected by *ync13-19* and/or *rng10Δ* mutants. (A–D)** Time course of DIC and middle focal plane of fluorescence images (left, in min), Bgs1 distribution (measured by FWHM) along the division site at the end of ring constriction (top right), and Bgs1 intensity at the division site over time (bottom right) in (A) WT, (B) *rng10Δ*, (C) *ync13-19*, and (D) *rng10Δ ync13-19* cells. One (C) or both daughter cells (D) lysed after cell separation. Cells were grown exponentially in YE5S + 1.2 M sorbitol liquid media at 25°C for 36 h, then washed and grown in YE5S without sorbitol at 36°C for 2 h before imaging at 36°C in imaging dish covered with YE5S agar medium. Thirteen slices spaced at 0.5 µm at each time point were taken every 2 min for 2 h. Bgs1 intensity at the division site was measured using sum intensity projections and the background subtracted. Line scans along the division site were fitted in Gaussian distribution to calculate FWHM (mean ± SD) for cells at the end of ring constriction (the single time point when the ring has constricted to a dot in cell center and Rlc1 has reached the highest pixel intensity). Bgs1 intensity profiles at the division site over time were plotted and cells were aligned using the end of Rlc1 ring constriction (A–C) or peak Bgs1 intensity (D). *P*-values were from Welch's *t* test. The underlying data (for panels A–D) can be found in S1 Raw Data file. Bars, 5 µm.
(TIF)

**S8 Fig. Trs120 localization at the division site in WT, *ync13Δ, rga7Δ, rng10Δ*, and *rga7Δ rng10Δ* cells and the correlation between Ync13 intensity and Ypt3 vesicles at the division site. (A)** Rlc1-tdTomato marks the position and diameter of the contractile ring. Trs120-3GFP accumulates inside the area with the ring in *ync13Δ* cells, but outside the ring area in *rga7Δ, rng10Δ*, and *rga7Δ rng10Δ* cells at the division site. The middle focal planes along with the sum intensity projection of the GFP channel from a 2-min continuous movie (exposure time 200 ms) without delay are shown. Arrowheads mark Trs120-3GFP in cells with constricting ring. Cells were grown exponentially in EMM5S liquid media for ~48 h before imaging. **(B)** Correlation between Ync13 intensity and the number of Ypt3 vesicles at the division site. (Left) Maximal intensity projection of cells expressing both *3nmt1-mECitrine-ync13* and *tdTomato-ypt3*. (Right) The number of Ypt3 vesicles arriving at the division site during the 2 min continuous movie vs. Ync13 intensity. Cells were grown in YE5S + thiamine liquid medium for 24 h and then grown in YE5S without thiamine for 16 h before imaging. The underlying data (for panel B) can be found in S1 Raw Data file. Bars, 5 µm.
(TIF)

**S1 Video. Movement of Trs120-3GFP vesicles to the division site in WT cells.** The movies track the movement of Trs120-3GFP vesicles from the cytoplasm to the division site in WT cells using colored lines. The videos correspond to Figs 5 and S8A. Strains were grown exponentially in EMM5S liquid media at 25°C for ~48 h before imaging in continuous movies without delay. Time, min:s.
(AVI)

**S2 Video. Movement of Trs120-3GFP vesicles to the division site in *rng10Δ* cells.** The movies track the movement of Trs120-3GFP vesicles from the cytoplasm to the division site in *rng10Δ* cells using colored lines. The videos correspond to

Figs 5 and S8A. Strains were grown exponentially in EMM5S liquid media at 25°C for ~48h before imaging in continuous movies without delay. Time, min:s.
(AVI)

**S3 Video. Movement of Trs120-3GFP vesicles to the division site in *ync13Δ* cells.** The movies track the movement of Trs120-3GFP vesicles from the cytoplasm to the division site in *ync13Δ* cells using colored lines. The videos correspond to Figs 5 and S8A. Strains were grown exponentially in EMM5S liquid media at 25°C for ~48h before imaging in continuous movies without delay. Time, min:s.
(AVI)

**S1 Table. Proteins identified in mass spectrometry of affinity-purified Ync13-3xFlag from *Schizosaccharomyces pombe*[a] .**
(DOCX)

**S2 Table. *Schizosaccharomyces pombe* strains used in this study.**
(DOCX)

**S1 Raw images. Uncropped Western blots/SDS–PAGE gels for figures.** Each of the full-size, uncropped, and labeled Western blot or SDS–PAGE gel is shown for the indicated figure panels. The relevant protein is labeled and indicated with an arrowhead. Irrelevant lanes, which are not included in figures, are labeled on the top.
(PDF)

**S2 Raw images. Uncropped representative electron microscopy images of *ync13Δ* cells shown in Fig 7B.**
(PDF)

**S1 Raw Data. This file contains all underlying data of the graphs shown in the manuscript.** Each sheet corresponds to one figure panel.
(XLSX)

## Acknowledgments

We thank Juan Carlos, Juan Ribas, and Yajun Liu for yeast strains; Jean Daraspe, Olivia Muriel, and the University of Lausanne Electron Microscopy Facility for help with electron microscopy; Anita Hopper, Jim Hopper, Steve Osmani, and Damien Wilburn for equipment; and current and former members of the Wu and Martin groups for helpful discussions.

## Author contributions

**Conceptualization:** Sha Zhang, Davinder Singh, Jian-Qiu Wu.

**Formal analysis:** Sha Zhang, Davinder Singh, Katherine J. Zhang, Jian-Qiu Wu.

**Funding acquisition:** Sha Zhang, Sophie G. Martin, Jian-Qiu Wu.

**Investigation:** Sha Zhang, Davinder Singh, Yi-Hua Zhu, Katherine J. Zhang, Alejandro Melero, Jian-Qiu Wu.

**Methodology:** Sha Zhang, Davinder Singh, Yi-Hua Zhu, Alejandro Melero, Jian-Qiu Wu.

**Resources:** Sophie G. Martin, Jian-Qiu Wu.

**Supervision:** Jian-Qiu Wu.

**Validation:** Sha Zhang, Davinder Singh.

**Visualization:** Sha Zhang, Davinder Singh, Jian-Qiu Wu.

**Writing – original draft:** Sha Zhang, Davinder Singh.

**Writing – review & editing:** Sha Zhang, Davinder Singh, Sophie G. Martin, Jian-Qiu Wu.

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
