## [Editor Report · Decision Letter 0]

19 Sep 2025

Dear Dr Wu,

Thank you for submitting your revised manuscript from Review Commons entitled "Spatial control of secretory vesicle targeting by the Ync13-Rga7-Rng10 complex during cytokinesis" for consideration as a Research Article by PLOS Biology.

Your manuscript has now been evaluated by the PLOS Biology editorial staff, as well as by an academic editor with relevant expertise, and I am writing to let you know that we would like to send your submission back out for re-review by the original reviewers at Review Commons.

Once your full submission is complete, your paper will undergo a series of checks in preparation for peer review. After your manuscript has passed the checks it will be sent out for review. To provide the metadata for your submission, please Login to Editorial Manager (https://www.editorialmanager.com/pbiology) within two working days, i.e. by Sep 21 2025 11:59PM.

Kind regards,

Richard

Richard Hodge, PhD

rhodge@plos.org

PLOS

---

## [Decision Letter · Decision Letter 1]

8 Oct 2025

Dear Dr Wu,

Thank you for your patience while we considered your revised manuscript from Review Commons entitled "Spatial control of secretory vesicle targeting by the Ync13-Rga7-Rng10 complex during cytokinesis" for publication as a Research Article at PLOS Biology. This revised version of your manuscript has been evaluated by the PLOS Biology editors, the Academic Editor and the original reviewers at Review Commons.

Based on the reviews, I am pleased to say that we are likely to accept this manuscript for publication, provided you satisfactorily address the remaining point raised by Reviewer #1. Please also make sure to address the following data and other policy-related requests that I have provided below (A-F):

(A) We routinely suggest changes to titles to ensure maximum accessibility for a broad, non-specialist readership. In this case, we would suggest a minor edit to the title, as follows. Please ensure you change both the manuscript file and the online submission system, as they need to match for final acceptance:

“The Ync13-Rga7-Rng10 complex selectively coordinates secretory vesicle trafficking and secondary septum formation during cytokinesis”

(B) Please paste the Funding statement provided in page 36 into the ‘Finanical Disclosure’ section of the online submission form. In addition, please replace the last sentence with the following declarative statement:

“The funders did not play any role in the study design, data collection and analysis, decision to publish, or preparation of the manuscript.”

(C) Thank you for already providing the underlying data for the following figures:

Figure 3B, 3F-G, 4D-E, 4G, 5B, 5D, 6C, 7C, S6A-C, S7A-D, S8B

I have checked the S1 Raw Data file and this all looks good. However, I just wanted to ask whether underlying data should be included for Figures 2E and/or S5F?

(D) Please also ensure that each of the relevant figure legends in your manuscript include information on *WHERE THE UNDERLYING DATA CAN BE FOUND*, and ensure your supplemental data file/s has a legend.

(E) Per journal policy, if you have generated any custom code during the course of this investigation, please make it available without restrictions. Please ensure that the code is sufficiently well documented and reusable, and that your Data Statement in the Editorial Manager submission system accurately describes where your code can be found.

(F) Please ensure that you are using best practice for statistical reporting and data presentation. These are our guidelines https://journals.plos.org/plosbiology/s/best-practices-in-research-reporting#loc-statistical-reporting and a useful resource on data presentation https://journals.plos.org/plosbiology/article?id=10.1371/journal.pbio.1002128

- If you are reporting experiments where n ≤ 5, please plot each individual data point.

We expect to receive your revised manuscript within two weeks.

*Published Peer Review History*

*Press*

Best regards,

Richard

Richard Hodge, PhD

rhodge@plos.org

Reviewer remarks:

Reviewer #1: The authors have responded satisfactorily to my comments and I have recommended acceptance. The paper is a very nice addition to the field.

The one minor point is that the numbers written in blue on the plates in Figure 6 are nearly impossible to see. I recommend that the numbers be added in a way that makes it easier to determine what is what. One possibility is to use a different color and larger numbers. Another is to add the numbers to the background outside of the plates.

Reviewer #2: The authors have addressed all my concerns. There is no additional comment from me.

Reviewer #3: Authors have done a good job for revision. I am satisfied with their revised manuscript.

---

## [Editor Report · Decision Letter 2]

14 Oct 2025

Dear Dr Wu,

On behalf of my colleagues and the Academic Editor, Anna Akhmanova, I am pleased to say that we can accept your manuscript for publication, provided you address any remaining formatting and reporting issues. These will be detailed in an email you should receive within 2-3 business days from our colleagues in the journal operations team; no action is required from you until then. Please note that we will not be able to formally accept your manuscript and schedule it for publication until you have completed any requested changes.

PRESS

Best wishes,

Richard 

Richard Hodge, PhD

rhodge@plos.org

PLOS
